# RAGE deficiency predisposes mice to virus-induced paucigranulocytic asthma

Jaisy Arikkatt[1†], Md Ashik Ullah[1,2†], Kirsty Renfree Short[1,3†], Vivan Zhang[1], Wan Jun Gan[1], Zhixuan Loh[1], Rhiannon B Werder[1], Jennifer Simpson[1], Peter D Sly[3,4], Stuart B Mazzone[1], Kirsten M Spann[3,5,6], Manuel AR Ferreira[7], John W Upham[3,8], Maria B Sukkar[2,9], Simon Phipps[1,3*]

[1]School of Biomedical Science, University of Queensland, Brisbane, Australia; [2]Woolcock Institute of Medical Research, Sydney Medical School, University of Sydney, New South Wales, Australia; [3]Australian Infectious Diseases Research Centre, The University of Queensland, Brisbane, Australia; [4]Centre for Children's Health Research Children's Health Queensland, The University of Queensland, Brisbane, Australia; [5]School of Chemistry and Molecular Biosciences, The University of Queensland, St. Lucia, Australia; [6]School of Biomedical Science, Queensland University of Technology, Brisbane, Australia; [7]QIMR Berghofer Medical Research Institute, Brisbane, Australia; [8]School of Medicine, The University of Queensland, Princess Alexandra Hospital Brisbane, Brisbane, Australia; [9]Discipline of Pharmacy, Graduate School of Health, University of Technology Sydney, Sydney, Australia

**Abstract** Asthma is a chronic inflammatory disease. Although many patients with asthma develop type-2 dominated eosinophilic inflammation, a number of individuals develop paucigranulocytic asthma, which occurs in the absence of eosinophilia or neutrophilia. The aetiology of paucigranulocytic asthma is unknown. However, both respiratory syncytial virus (RSV) infection and mutations in the receptor for advanced glycation endproducts (*RAGE*) are risk factors for asthma development. Here, we show that RAGE deficiency impairs anti-viral immunity during an early-life infection with pneumonia virus of mice (PVM; a murine analogue of RSV). The elevated viral load was associated with the release of high mobility group box-1 (HMGB1) which triggered airway smooth muscle remodelling in early-life. Re-infection with PVM in later-life induced many of the cardinal features of asthma in the absence of eosinophilic or neutrophilic inflammation. Anti-HMGB1 mitigated both early-life viral disease and asthma-like features, highlighting HMGB1 as a possible novel therapeutic target.

*For correspondence: s.phipps@uq.edu.au

†These authors contributed equally to this work

Competing interests: The authors declare that no competing interests exist.

## Introduction

Asthma is a chronic inflammatory disease of the airways that affects more than 300 million people worldwide (*Pawankar, 2014*). The asthmatic airway is characterised by numerous structural changes (collectively referred to as 'airway remodelling') including increased airway smooth muscle (ASM) mass, sub-epithelial fibrosis and goblet cell hyperplasia/hypersecretion. Asthma is often considered to be driven by type-2 eosinophilic inflammation (*Lambrecht and Hammad, 2003*). However, approximately half of all patients with asthma do not exhibit eosinophilic inflammation, and accordingly, other asthma phenotypes have been described that include neutrophilic, mixed granulocytic and paucigranulocytic inflammatory phenotypes (*Simpson et al., 2006*;*Hinks et al., 2016*). Paucigranulocytic asthma occurs in the absence of pronounced eosinophilia or neutrophilia and in the few studies performed to date, is reported to affect between 32% and 52% of patients with asthma

(*Simpson et al., 2006*; *Schleich et al., 2013*; *Simpson et al., 2007*; *Porsbjerg et al., 2009*; *Wang et al., 2011*). At present, there is limited information available regarding the stability of this phenotype and little is known about the pathogenic mechanisms or 'endotype' that underlies disease development.

The majority of asthma cases begin in early childhood. Interestingly, O'Reilly and colleagues (*O'Reilly et al., 2013*) showed that wheezy preschool children displayed ASM remodelling prior to their diagnosis of asthma as school aged children. Wheeze in children is often associated with a lower respiratory tract infection, raising the possibility that paediatric viral infections contribute to airway remodelling in early life and the subsequent development of asthma in later life. Consistent with this notion, epidemiological studies have documented an association between severe respiratory syncytial virus (RSV) infection early in life and asthma development (*James et al., 2013*). Indeed, persistent RSV-induced wheezing before 3 years of age is associated with chronic asthma persisting into adulthood (*Taussig et al., 1989*; *Wright et al., 1989*). However, whether early life viral infections contribute to the development of paucigranulocytic asthma has yet to be investigated.

In addition to viral infections, specific immune signaling pathways have also been implicated in the development of asthma. For example, we have previously shown that $Tlr7^{-/-}$ mice display the cardinal features of asthma following an earlier life and later life infection with pneumonia virus of mice (PVM; a murine analogue of RSV) (*Kaiko et al., 2013*). This is consistent with the observation that, compared to non-asthmatic individuals, TLR7 function is reduced in adolescents with asthma (*Roponen et al., 2010*). There is also a growing body of evidence implicating high mobility group box 1 (HMGB1) and the receptor for advanced glycation endproducts (RAGE) in disease development (*Sukkar et al., 2012*). HMGB1 is an alarmin that acts as a non-histone nuclear protein in quiescent cells. Upon cell stimulation (such as by the presence of a pathogen) HMGB1 becomes phosphorylated and is translocated from the nucleus to the cytoplasm. Upon release, extracellular HMGB1 can bind to receptors such as RAGE, toll-like receptor (TLR) two and TLR4, and trigger a pro-inflammatory response (*Lotze and Tracey, 2005*). Increased levels of HMGB1 have been detected in the sputum of patients with asthma and high levels of HMGB1 have been associated with reduced lung function (*Hou et al., 2011*).

The contribution of RAGE to asthma risk and symptoms is less well understood. This receptor is expressed as membrane-bound (mRAGE) and soluble (sRAGE) forms; the latter is produced primarily by shedding of mRAGE (*Raucci et al., 2008*) and is thought to function as a 'decoy' receptor, preventing RAGE ligands from interacting with mRAGE, which would normally trigger inflammation (*Schmidt, 2015*). Of note, a non-synonymous variant (rs2070600) in the *AGER* gene (which encodes RAGE) is a strong determinant of serum sRAGE levels (*Jang et al., 2007*; *Maruthur et al., 2015*), with the rs2070600:C allele (encoding a Gly at position 82) being associated with increased sRAGE. This same allele is associated with decreased risk of allergic asthma (OR = 0.87, p=0.003 in [*Ferreira et al., 2014*]). Taken together, the results from these genetic studies indicate that decreased signalling through mRAGE (as a result of increased sRAGE production) might have a protective effect in allergic asthma. This prediction is complicated by the apparently conflicting observation that rs2070600:C (which results in increased sRAGE) is associated with decreased FEV1/FVC ratio. However, it is important to note that this analysis was performed in the general population (*Repapi et al., 2010*), and hence cannot be associated with asthma or allergic disease. Intriguingly, RAGE deficiency or HMGB1 neutralisation ameliorates type-2 inflammation in preclinical models of allergic asthma (*Ullah et al., 2014*). However, whether RAGE or HMGB1 contribute to viral bronchiolitis in early life, and the subsequent development of viral-induced asthma, remains to be determined.

Here, using RAGE deficient mice and PVM infection we show that RAGE deficiency severely impairs antiviral immunity. The resulting increase in viral load in the airway epithelium led to the release of HMGB1, which via the activation of TLR4 was instrumental in driving ASM remodelling during an early life viral infection, as well as the development of mucus hypersecretion and airway hyperreactivity in later life. Critically, these features occurred independently of type-2 inflammation and granulocytic inflammation, suggesting that blocking HMGB1 may be a promising therapeutic approach to prevent the development of airway remodelling, which can occur in the absence of eosinophilic or neutrophilic inflammation.

## Results

### RAGE deficiency results in increased viral infection and reduced anti-viral immunity during a PVM infection in early life

To understand the role of RAGE in an early life respiratory viral infection, seven-day old RAGE-deficient (RAGE⁻) and wild-type (WT) mice were inoculated with PVM and viral infection was assessed by qPCR at seven days post-infection (dpi). Day seven was selected as the time point of interest, as this represents the peak of PVM replication in mice (*Kaiko et al., 2013*; *Bonville et al., 2006*). Infected RAGE deficient mice had significantly higher levels of PVM RNA seven dpi compared to infected WT mice (*Figure 1A*). However, in light of the fact that qPCR cannot differentiate between infectious and non-infectious viral particles, and the fact that PVM titres are notoriously difficult to assess by plaque assay (*Dyer et al., 2007*), we also measured active PVM infection by immunohistochemistry. RAGE deficient mice displayed a significantly higher percentage of PVM infected airway epithelial cells (AECs) in the respiratory tract compared to WT mice at seven dpi (*Figure 1B & C*). This increased infection was associated with lower levels of IFNα, IFNλ and IFNγ in the BAL fluid and lungs at seven dpi in PVM-infected RAGE deficient mice (*Figure 1D* & *Figure 1E*). Interestingly, there was no significant difference in IFN production between mock- and PVM-infected RAGE deficient mice, suggesting that RAGE deficient mice were unable to mount an IFN response following infection with PVM (p>0.999). Consistent with this notion, PVM-infected RAGE deficient mice had significantly lower expression of interferon regulatory factor (*IRF7*) (which regulates IFNα production) relative to WT mice (*Figure 1E*).

pDCs are major producers of IFNα during infection (*Tailor et al., 2006*) and previous studies have demonstrated that RAGE plays a key role in the mobilisation of these cells in mice (*Manfredi et al., 2008*). We thus reasoned that the attenuated IFN production and associated increased viral infection observed in RAGE deficient mice would be associated with defective pDC recruitment. To determine whether lung pDCs expressed RAGE, we took advantage of the RAGE deficient mouse in which the functional *Ager* gene is replaced by *GFP*. Using flow cytometry, we demonstrated that pDCs derived from RAGE deficient mice were GFP⁺, indicating that under normal circumstances, wild-type lung pDCs express RAGE (*Figure 1F*). We next determined whether the absence of RAGE would affect pDC recruitment to the lungs. At seven dpi, there were significantly fewer pDCs in the lungs of PVM-infected RAGE deficient mice compared to PVM-infected WT mice (*Figure 1G*), and similar to the IFN response, there was no significant difference in pDC recruitment between mock- and PVM-infected RAGE deficient mice (*Figure 1G*) (p=0.6769). Together, these data demonstrate that reduced IFN production in PVM-infected RAGE deficient mice is associated with a defect in pDC recruitment.

### The absence of RAGE increases HMGB1 levels and ASM mass during PVM infection in early life

Respiratory viral infections induce the production of various alarmins which may play an important role in the development of asthma (*Kaiko et al., 2013*; *You et al., 2015*; *Lloyd, 2010*). Thus, we measured the levels of the alarmins IL-33, IL-1α and HMGB1 in the lungs of neonatal WT and RAGE deficient mice at seven dpi. Whilst the levels of IL-33 and IL-1α were elevated in infected WT mice, there was no difference in expression between mock- and PVM-infected RAGE deficient mice (*Figure 2A*) (p>0.999). In contrast, PVM-infected RAGE deficient mice had significantly higher levels of HMGB1 in the BAL fluid compared to both mock-infected RAGE deficient mice and PVM-infected WT mice (*Figure 2A*). PVM-infected RAGE deficient mice also had significantly higher levels of cytoplasmic HMGB1 in AECs compared to the other treatment groups (*Figure 2B*). These findings suggest that loss of RAGE leads to a dramatic increase in HMGB1 release and that AECs are likely to be an important source of this alarmin.

ASM remodelling is a hallmark feature of asthma that can commence in early life (*O'Reilly et al., 2013*). Since HMGB1 has been implicated in smooth muscle proliferation (*Porto et al., 2006*), we hypothesised that the elevated HMGB1 would induce ASM remodelling in early life. Indeed, there was significantly more ASM mass in PVM-infected RAGE deficient mice compared to mock-infected RAGE deficient mice and PVM-infected WT mice (*Figure 2C*). In contrast, PVM-infected WT mice did not exhibit any increase in ASM growth compared to their mock-infected counterparts

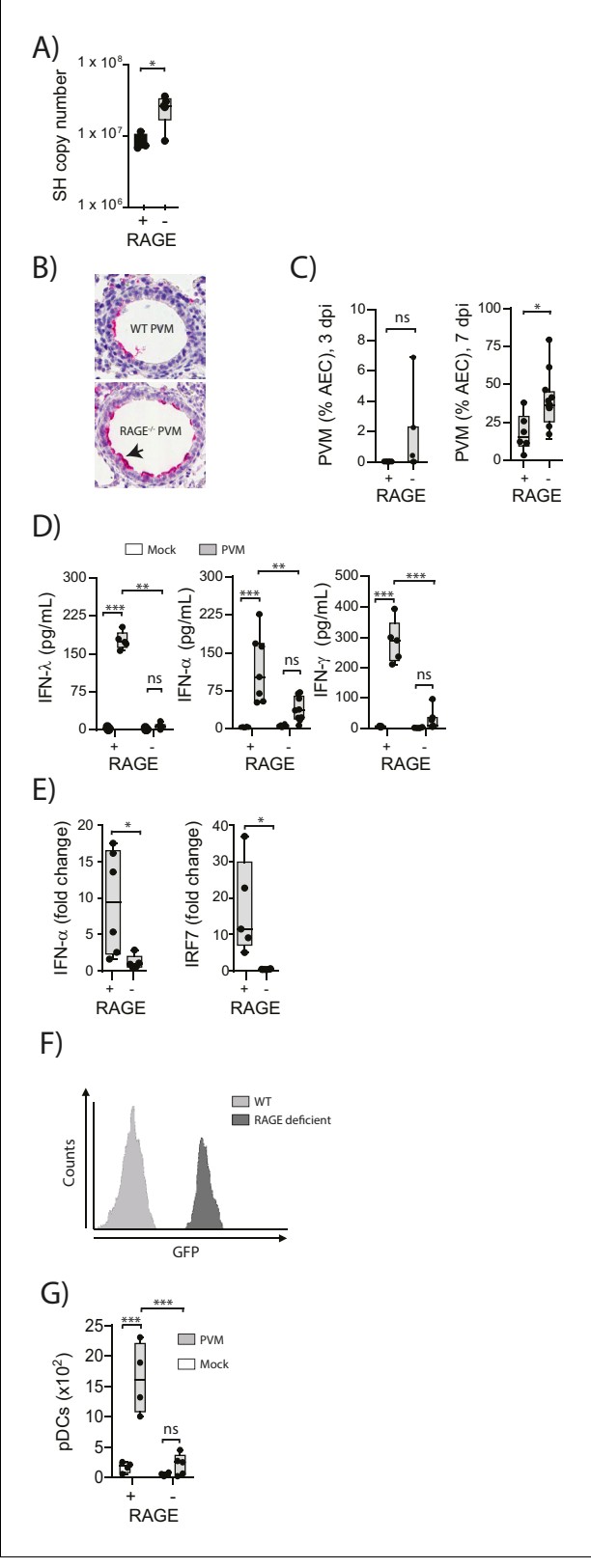

**Figure 1.** Increased virus replication and defective innate immunity in PVM-infected neonatal RAGE deficient mice. (A) RAGE deficient (RAGE⁻) and WT (RAGE⁺) mice were infected with 10 PFU of PVM at seven days of age and the viral load in the lungs (as determined by qPCR) was assessed 7 days post-infection (dpi). (B) Representative images of PVM staining in the lungs of WT and RAGE deficient mice seven dpi. Arrow indicates positive staining. (C)

*Figure 1 continued*

Percentage of airway epithelial cells (AECS) positive for PVM by immunohistochemistry. (D) Interferon levels in the bronchoalveolar lavage fluid of neonatal mice seven days post-PVM or mock infection (i.e. at 14-days of age). (E) The expression of *IFN-α4* and *IRF7* in the lungs of RAGE deficient and WT mice seven dpi. The expression of *IFN-α4* and *IRF7* was determined using the *ΔΔCt* method where naïve WT and RAGE deficient mice were used as the calibrator samples. (F) Histogram demonstrating GFP expression by plasmacytoid dendritic cells (pDCs) derived from naïve adult RAGE deficient and WT mice. pDCs were defined as CD11c+CD11b-B220+SiglecH+CD45RA+ cells. (G) The number of pDCs presents in the lungs of neonatal mice seven days post-PVM or mock infection (i.e. at 14-days of age). Statistical significance was determined by a Student's t-test (A, C and E) or a two-way ANOVA with Tukey's multiple comparisons test (D and G). Statistical significance is denoted by asterisks (*p<0.05;**p<0.01; ***p<0.001; ns: not significant). Box and whisker plots show the minimum value, the median and the maximum value. Data are representative of two independent experiments. n = 3–10.

(*Figure 2C*) (p>0.9999). Thus, in the absence of RAGE, an early life PVM infection is associated with increased levels of HMGB1 and ASM mass.

## Neutralising HMGB1 reduces PVM infection and ASM mass in RAGE deficient mice during an early life PVM infection

In light of the association observed between extracellular HMGB1 levels and ASM mass, we next investigated whether neutralisation of HMGB1 would prevent the induction of ASM remodelling. Daily treatment of PVM-infected RAGE deficient mice with anti-HMGB1 antibody (50 μg) from 2 to 6 dpi significantly decreased the levels of extracellular HMGB1 in the BAL fluid (*Figure 3A and B*). The number of AECs positive for cytoplasmic HMGB1 was also significantly decreased following anti-HMGB1 treatment (*Figure 3B*). Importantly, ASM mass was significantly lower in RAGE deficient mice that received anti-HMGB1 compared to isotype control antibody (*Figure 3C*). Anti-HMGB1 also significantly reduced the viral infection in the lungs of RAGE deficient mice (*Figure 3D*), and significantly increased IFNα4 and IFNγ expression in the lung compared to mice treated with the isotype control (*Figure 3E*). In contrast, anti-HMGB1 treatment did not affect viral load or ASM mass in PVM-infected WT mice (*Figure 4A–C*) (p=0.1616 and p=0.5109). Together, these data demonstrate that in the absence of RAGE, PVM infection increases ASM mass in an HMGB1 dependent manner.

## RAGE/TLR4 deficient mice do not display increased ASM mass upon an early life PVM infection

We next sought to identify the receptor through which HMGB1 signalled to trigger ASM remodelling in RAGE deficient mice. As TLR4 is a known receptor for HMGB1 (*Lotze and Tracey, 2005*; *Ullah et al., 2014*), we investigated viral load and ASM mass in mice deficient in both RAGE and TLR4. There was no significant difference in viral load between PVM-infected RAGE/TLR4 deficient mice and PVM-infected RAGE deficient mice at seven dpi (*Figure 5A*) (p=0.1693). However, ASM growth was attenuated in RAGE/TLR4 deficient mice compared to RAGE deficient mice (*Figure 5B*). TLR4 deficiency alone had no significant effect on either viral load or ASM mass when compared to WT mice (*Figure 5A and B*) (p>0.9999).

## Administration of exogenous IFN-α limits viral replication, ASM remodelling and HMGB1 release in RAGE deficient mice during an earl life PVM infection

The above data suggested that in the absence of RAGE, PVM-induced pDC recruitment is impaired and there is decreased interferon production and increased viral infection. This is associated with increased HMGB1 levels which then drives an increase in ASM mass. This model would thus suggest that the addition of exogenous interferon would not only reduce virus replication in RAGE deficient mice but that it would also reduce the levels of extracellular HMGB1 and ASM mass. In order to test this hypothesis, neonatal RAGE deficient mice were treated with 5000 U of IFN-α or diluent alone at three dpi. At seven dpi, the mice were euthanised and the viral load, HMGB1 levels and ASM mass were assessed (*Figure 6A*). Exogenous IFN-α significantly decreased viral load in AECs compared to mock-treated mice (*Figure 6B*). Moreover, both cytoplasmic and extracellular HMGB1 were

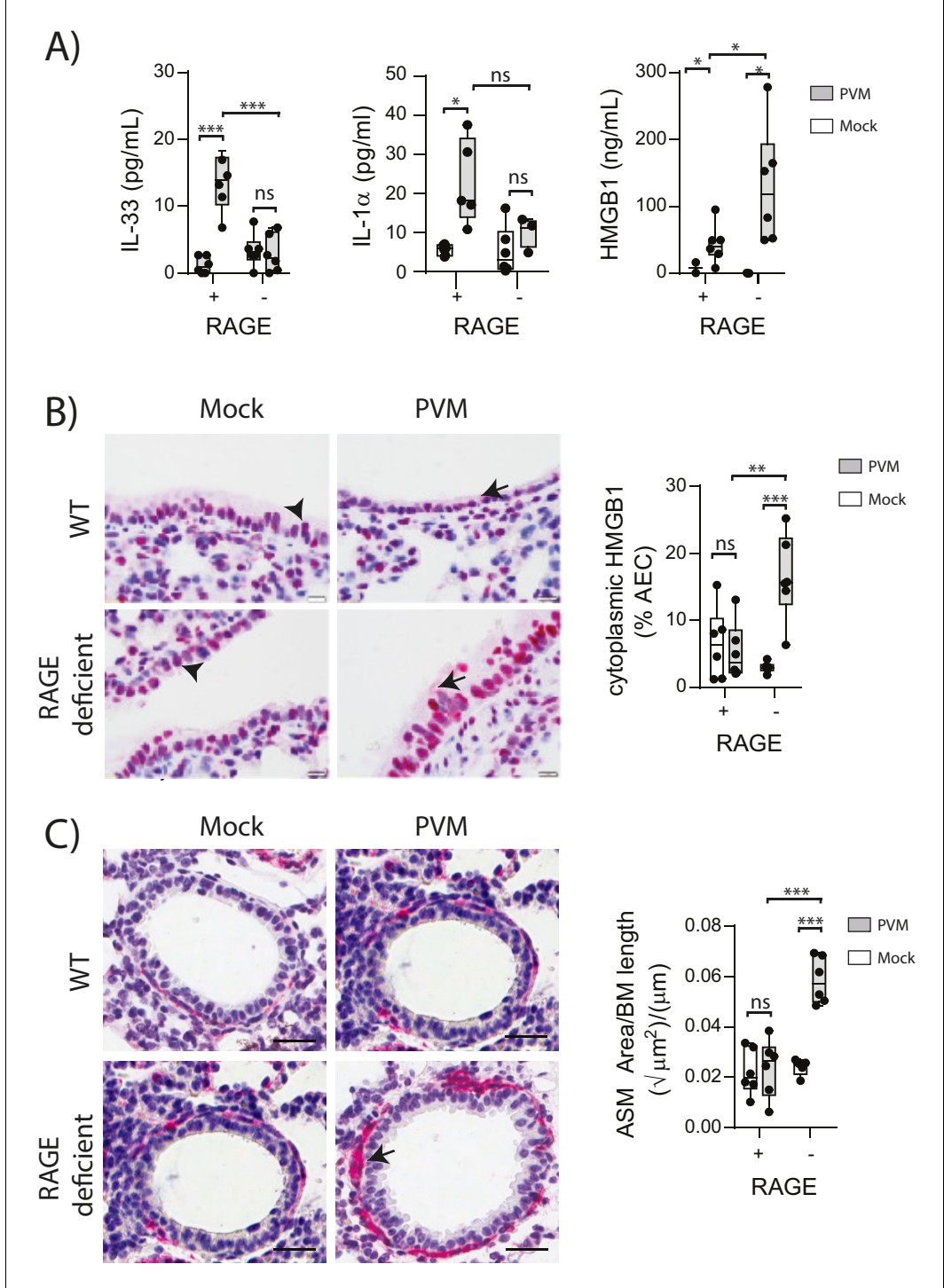

**Figure 2.** PVM-infected neonatal RAGE deficient mice have higher levels of cytoplasmic and extracellular HMGB1. (**A**) Levels of alarmins in the bronchoalveolar lavage fluid of neonatal mice. RAGE deficient (RAGE⁻) and WT (RAGE⁺) mice were infected with 10 PFU of PVM or mock at 7 days of age and alarmin levels were assessed seven dpi. (**B**) Representative images of mouse lung tissue sections stained for HMGB1 seven dpi (i.e. at 14-days of age). Original magnification x400 and scale bars indicate 20 μm. HMGB1 staining is shown in red whilst the nucleus is stained in blue. Arrows indicate cytoplasmic HMGB1, arrowheads indicate nuclear HMGB1. The percentage of airway epithelial cells (AECs) with cytoplasmic HMGB1 was quantified as described in the materials and methods. (**C**) Representative images of mouse lung tissue sections stained for smooth muscle actin seven dpi (i.e. at 14-days of age). Original magnification x400 and scale bars indicate 20 μm. Smooth muscle staining is shown in red whilst
*Figure 2 continued on next page*

*Figure 2 continued*

the nucleus is stained in blue. Arrow indicates positive staining. The airway smooth muscle area (ASM) was calculated relative to the basement membrane (BM) length of small airways. Statistical significance was determined by a two-way ANOVA with Tukey's multiple comparisons test. Statistical significance is denoted by asterisks (*p<0.05;**p<0.01; ***p<0.001; ns: not significant). Box and whisker plots show the minimum value, the median and the maximum value. Data are representative of two independent experiments. n = 3–6.

significantly decreased in IFN-treated mice, and this was associated with a significant reduction in ASM growth (*Figure 6C and D*).

## PVM does not induce a pronounced type-2 response in neonatal RAGE deficient mice

We have previously shown that TLR7 deficiency predisposes mice to severe PVM bronchiolitis and an IFN-$\alpha^{low}$, IL-33$^{high}$ cytokine microenvironment (*Kaiko et al., 2013*). This response then leads to the onset of type-2 inflammation early in life which, upon re-infection in later life, is re-activated to promote the hallmark pathologies of asthma (*Kaiko et al., 2013*). Therefore, we next investigated the inflammatory profile of RAGE deficient and WT mice following mock or PVM infection. Analysis of the BAL cells by differential cell count showed that there was no significant difference in the numbers of eosinophils, neutrophils or lymphocytes between mock- and PVM-infected RAGE deficient mice (*Figure 7A*) (p=0.1161; p=0.6107; p=0.0826 and p=0.8023). Consistent with the lack of an eosinophilic response, IL-5 and IL-13 levels were not significantly elevated in the PVM-infected RAGE deficient mice (*Figure 7B*) (p=0.7016 and p=0.9973) Levels of IL-17A were also not elevated in RAGE deficient mice upon infection with PVM (*Figure 7B*). A similar trend was observed for eotaxin-2, CXCL1 (KC), IL-12p40, IL-1$\beta$, IL-6, IL-23p19 and CCL3 (MIP-1$\alpha$) levels in lung homogenates (*Figure 7C*). Indeed, the only cytokine that was elevated in RAGE deficient mice upon infection with PVM was TNF-$\alpha$ (*Figure 7C*). In order to confirm that the observed phenotype occurred in the absence of a pronounced type-2 response, PVM-infected RAGE deficient mice were treated with soluble IL-13R$\alpha$2 fusion protein (to neutralise IL-13) (*Figure 8A*). Blocking IL-13 failed to reduce viral infection of AECs, ASM mass, or cytoplasmic HMGB1 levels in AECs (*Figure 8B–D*). (p=0.7422; p=0.8369 and p=0.4266). Taken together, these data indicate the absence of a type-2 inflammatory signature in neonatal RAGE deficient mice during PVM infection.

## Re-infection of RAGE deficient mice with PVM induces the features of asthma

In light of the ASM growth observed in neonatal RAGE deficient mice, we next questioned whether a secondary PVM infection would promote features of airway wall remodelling characteristic of asthma (*Figure 9A*). In contrast to the primary infection, viral antigen could not be detected in the lungs of reinfected RAGE deficient mice at various time points post-infection, presumably as a consequence of protective immunity (*Figure 9B*). Despite this, there was a significant increase in mucus-secreting cells in re-infected RAGE deficient mice compared to re-infected WT mice (*Figure 9C*). Re-infected RAGE deficient mice also displayed significantly higher levels of cytoplasmic HMGB1 in AECs and extracellular HMGB1 compared to their WT counterparts (*Figure 9D*). Consistent with this phenotype, re-infected RAGE deficient mice had more ASM mass compared to re-infected WT mice (*Figure 9E*) and increased airway resistance compared to all other treatment groups (*Figure 9F*). The number of ASM cells immunopositive for proliferating cell nuclear antigen (PCNA; a marker of active cell replication) was also significantly increased in re-infected RAGE deficient mice at three dpi compared to re-infected WT mice (*Figure 10A and B*). Taken together, these data indicated that in the absence of RAGE, re-infection with PVM induces some of the pathophysiological features of asthma.

## RAGE deficiency and secondary PVM infection predispose towards a paucigranulocytic asthma-like phenotype

In light of the goblet cell hyperplasia, airway hyperreactivity and increased ASM mass, we next questioned whether re-infected RAGE deficient mice had developed a type-2 response as we have

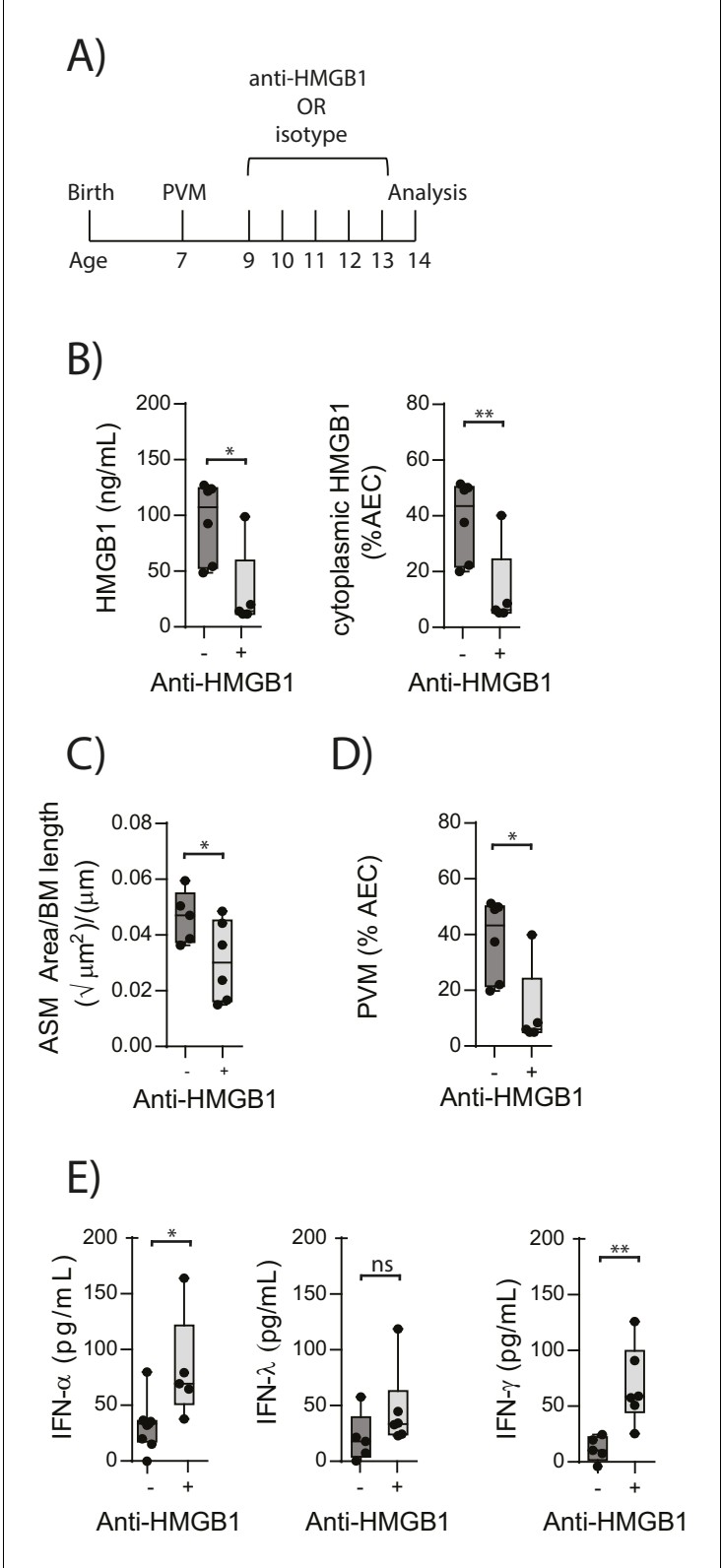

**Figure 3.** Blocking HMGB1 reduces ASM mass and virus replication in neonatal, PVM-infected RAGE deficient mice. (**A**) Schematic representation of the experimental protocol. Neonatal RAGE deficient mice were infected with 10 PFU of PVM at 7 days of age and treated with 50 μg of anti-HMGB1 or an IgG isotype control once per day from two to six days post-infection. (**B**) Levels of extracellular HMGB1 in the bronchoalveolar lavage fluid of

*Figure 3 continued on next page*

*Figure 3 continued*

treated mice seven days post-PVM infection. The percentage of airway epithelial cells (AECs) with cytoplasmic HMGB1 in treated mice is also displayed seven days post-PVM infection. (C) Lung sections were obtained from treated mice seven days post-PVM infection (i.e. at 14-days of age). These were then stained for smooth muscle actin by immunohistochemistry and the airway smooth muscle area (ASM) was calculated relative to the basement membrane (BM) length of small airways. (D) Percentage of airway epithelial cells (AECS) positive for PVM in treated mice seven days post-PVM infection (i.e. at 14-days of age). (E) Interferon levels in the bronchoalveolar lavage fluid of treated mice seven days post-PVM infection (i.e. at 14-days of age). Statistical significance was determined by a Student's t-test. Statistical significance is denoted by asterisks (*p<0.05;**p<0.01; ***p<0.001; ns: not significant). Box and whisker plots show the minimum value, the median and the maximum value. Data are derived from a single experiment. n = 5–7.

previously observed in TLR7 deficient mice (*Kaiko et al., 2013*). Similar to the primary PVM infection, the inflammatory response in the BAL (defined by the number eosinophils, neutrophils, mononuclear cells and lymphocytes) of re-infected RAGE deficient mice was comparable to WT mice and markedly lower than that of re-infected TLR7 deficient mice(*Figure 11A*). Similarly, IL-5, IL-13, periostin (a marker of type-2 inflammation) (*Jia et al., 2012*) and eotaxin-2 were not increased in RAGE deficient mice upon re-infection with PVM (*Figure 11B & C*) (p=0.8815, p>0.9999 and p>0.9999). Whilst an increase in CXCL1(KC) was observed in the lungs upon re-infection in RAGE deficient mice, this response was significantly lower than that of WT mice (*Figure 11C*). Therefore, the features of asthma observed in re-infected RAGE deficient mice resembled what is described in humans as paucigranulocytic asthma.

## Neutralising HMGB1 in re-infected RAGE deficient mice prevents the cardinal features of asthma

HMGB1 played an important role in driving ASM remodelling during the primary PVM infection. Therefore, we next assessed whether neutralising HMGB1 would protect RAGE deficient mice from developing the features of asthma following re-infection with PVM. RAGE deficient mice were

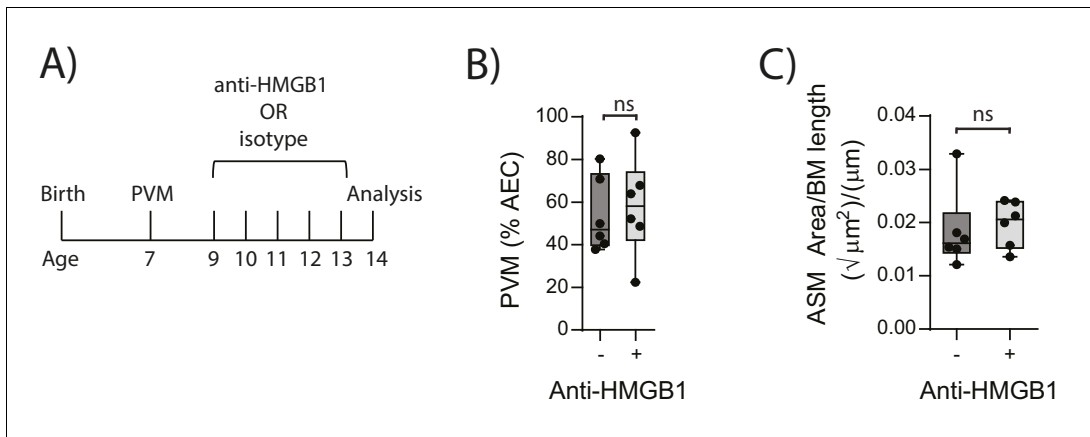

**Figure 4.** Blocking HMGB1 does not reduce ASM mass and virus replication in neonatal, PVM-infected WT mice. (A) Schematic representation of the experimental protocol. Neonatal WT mice were infected with 10 PFU of PVM at seven days of age and treated with 50 µg of anti-HMGB1 or an IgG isotype control once per day from two to six days post-infection. (B) Percentage of airway epithelial cells (AECS) positive for PVM in treated mice seven days post-PVM infection (i.e. at 14-days of age). (C) Lung sections were obtained from treated mice seven days post-PVM infection (i.e. at 14-days of age). These were then stained from smooth muscle actin by immunohistochemistry and the airway smooth muscle area (ASM) was calculated relative to the basement membrane (BM) length of small airways. Statistical significance was determined by a Student's t-test. Statistical significance is denoted by asterisks (*p<0.05;**p<0.01; ***p<0.001; ns: not significant). Box and whisker plots show the minimum value, the median and the maximum value. Data are derived from a single experiment. n = 6.

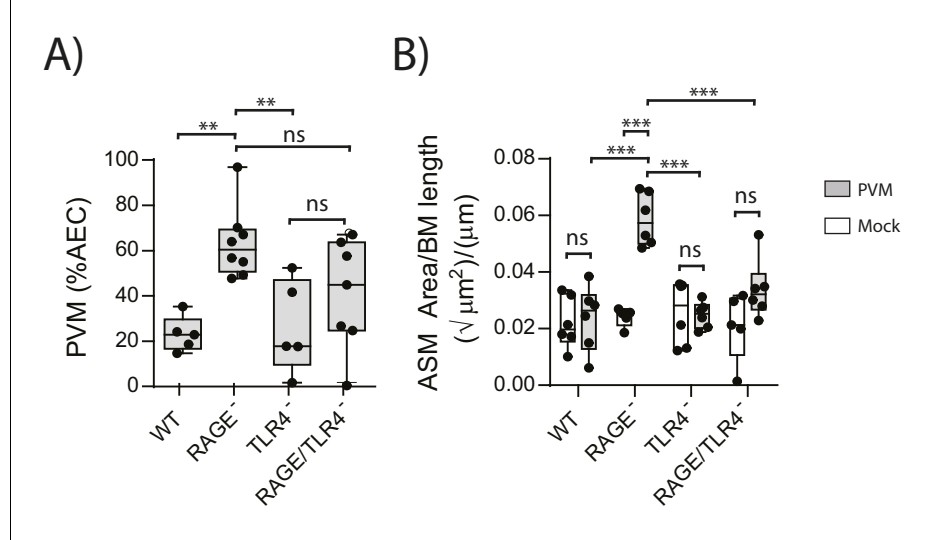

**Figure 5.** ASM mass is reduced in PVM-infected RAGE/TLR4 deficient mice. (**A**) Percentage of airway epithelial cells (AECS) positive for PVM by immunohistochemistry. RAGE deficient, WT (RAGE[+]), TLR4 deficient and RAGE/TLR4 deficient mice were infected with 10 PFU of PVM at 7 days of age and the viral infection was assessed seven days post-infection. (**B**) Lung sections were obtained from neonatal RAGE deficient, WT, TLR4 deficient and RAGE/TLR4 deficient mice seven days post-PVM infection (i.e. at 14-days of age). These were then stained from smooth muscle actin by immunohistochemistry and the airway smooth muscle area (ASM) was calculated relative to the basement membrane (BM) length of small airways. Statistical significance was determined by a two-way ANOVA with Tukey's multiple comparisons test. Statistical significance is denoted by asterisks (*p<0.05;**p<0.01; ***p<0.001; ns: not significant). Box and whisker plots show the minimum value, the median and the maximum value. Data are representative of two independent experiments. n = 5–8.

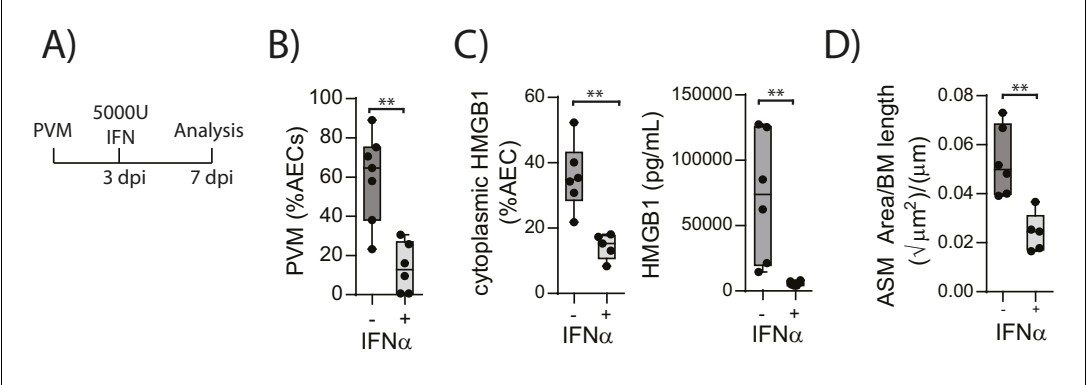

**Figure 6.** Interferon supplementation reduces virus replication, levels of extracellular HMGB1 and ASM mass and in neonatal, PVM-infected RAGE deficient mice. (**A**) Schematic representation of the experimental protocol. Neonatal RAGE deficient mice were infected with 10 PFU of PVM at seven days of age. Three days post-infection 5000 U of IFN-α or diluent only was administered intranasally to the mice. (**B**) Percentage of airway epithelial cells (AECS) positive for PVM in treated mice seven days post-PVM infection (i.e. at 14-days of age). (**C**) The percentage of airway epithelial cells (AECs) with cytoplasmic HMGB1 seven days post-PVM infection in treated mice. Levels of extracellular HMGB1 in the bronchoalveolar lavage fluid of treated mice seven days post-PVM infection is also displayed. (**D**) Lung sections were obtained from treated mice seven days post-PVM infection (i.e. at 14-days of age). These were then stained for smooth muscle actin by immunohistochemistry and the airway smooth muscle area (ASM) was calculated relative to the basement membrane (BM) length of small airways. Statistical significance was determined by a Student's t-test. Statistical significance is denoted by asterisks (*p<0.05;**p<0.01; ***p<0.001; ns: not significant). Box and whisker plots show the minimum value, the median and the maximum value. Data are representative of two independent experiments. n = 5–7.

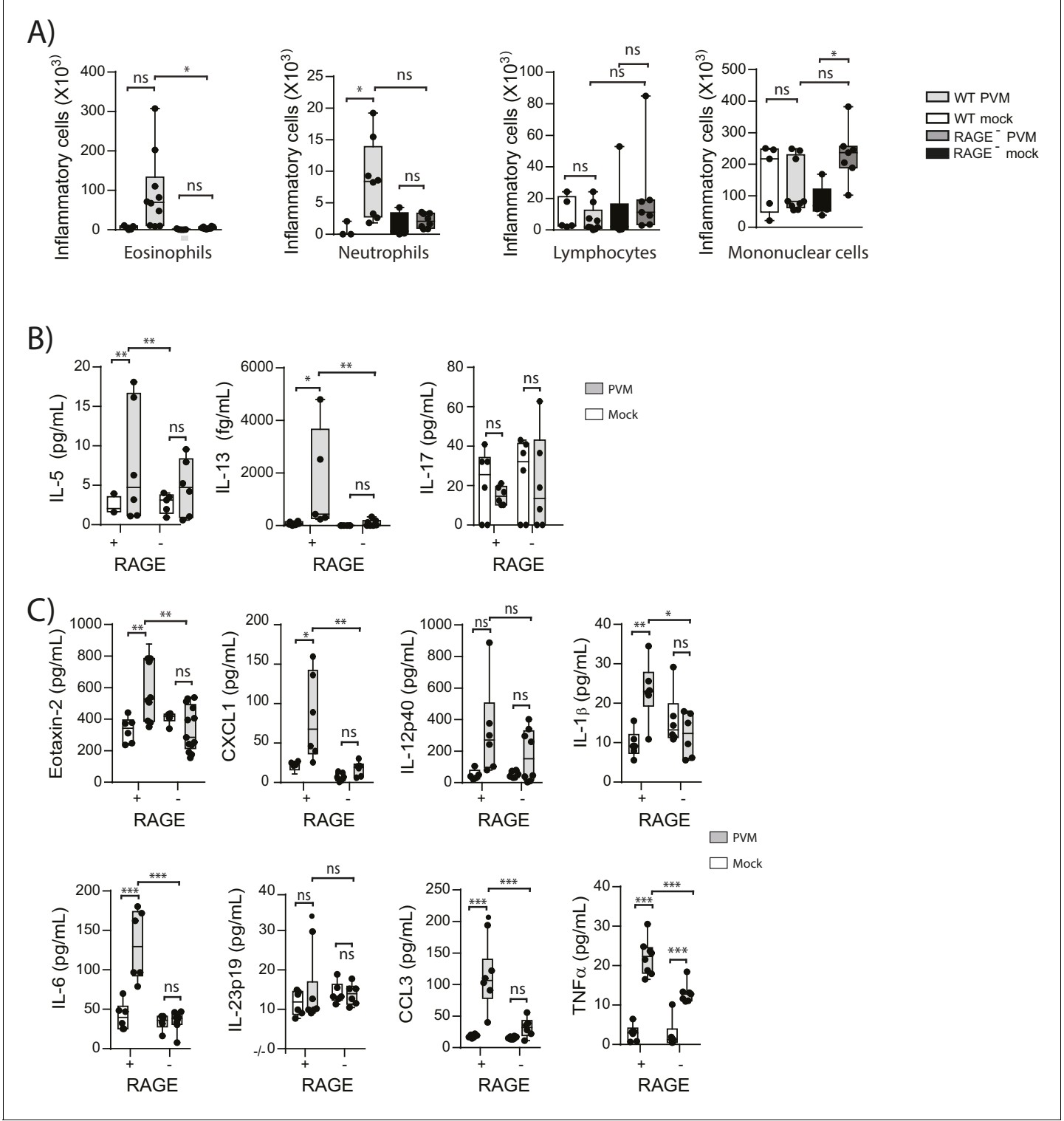

**Figure 7.** Neonatal RAGE deficient mice do not display a pronounced T$_H$2 response to an early life infection with PVM. RAGE deficient and WT (RAGE$^+$) mice were infected with 10 PFU of PVM or mock at seven days of age and the inflammatory profile was assessed seven days post-infection (**A**) Inflammatory cells in the bronchoalveolar lavage fluid were enumerated by differential counting following Geimsa staining. (**B**) Cytokine levels in the bronchoalveolar lavage fluid of neonatal mice seven days post-PVM or mock infection (i.e. at 14-days of age). (**C**) Cytokine levels in the lungs of neonatal mice seven days post-PVM or mock infection (i.e. at 14-days of age). Statistical significance was determined by a one-way ANOVA with Tukey's multiple comparisons test (**A**) or a two-way ANOVA (**B** and **C**). Statistical significance is denoted by asterisks (*p<0.05;**p<0.01; ***p<0.001; ns: not *Figure 7 continued on next page*

*Figure 7 continued*
significant). Box and whisker plots show the minimum value, the median and the maximum value. Data are representative two independent
experiments. n = 5–10.

treated daily for four days from the time of re-infection with anti-HMGB1 or an isotype-matched control (*Figure 12A*). Anti-HMGB1 significantly decreased the levels of cytoplasmic HMGB1 in AECs at seven dpi compared to treatment with the isotype control (*Figure 12B*). Importantly, anti-HMGB1 treated RAGE deficient mice also displayed significantly reduced goblet cell hyperplasia, ASM mass and AHR compared to control treated RAGE deficient mice (*Figure 12C–E*).

### RAGE/TLR4 deficient mice are protected from developing the cardinal features of asthma upon re-infection with PVM

Given that in the primary infection the increase in ASM mass was TLR4 dependent, we next sought to investigate if TLR4 in later life contributed to the development goblet cell hyperplasia, ASM mass and AHR. Thus, WT, RAGE deficient, TLR4 deficient and RAGE/TLR4 deficient mice were inoculated with PVM at seven days of age and later re-infected with mock or PVM 42fi days after the primary infection. Re-infected RAGE/TLR4 deficient mice had significantly fewer mucus secreting cells in the lung and displayed reduced ASM mass seven dpi compared to re-infected RAGE deficient mice (*Figure 13A and B*). Although there was no significant difference in AHR between the two mouse strains (p>0.9999), there was a trend towards decreased AHR in re-infected RAGE/TLR4 deficient mice (*Figure 13C*). To determine if the features of airway remodelling persisted, mice were re-assessed 21 days after re-infection. Goblet cell hyperplasia, ASM mass and AHR were elevated in RAGE deficient mice compared to WT mice (*Figure 13D–F*). Once again, these features were absent in RAGE/TLR4 deficient mice (*Figure 13D–F*), suggesting that TLR4 plays an important role in the development of the observed asthma phenotype.

### Discussion

Multiple genetic and environmental factors are implicated in the pathogenesis of asthma , and these gene-environment interactions most likely underpin the different endotypes that give rise to

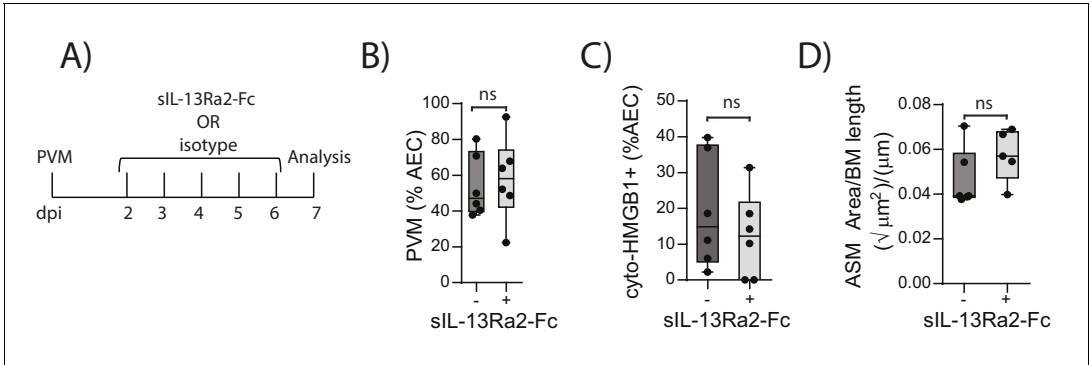

**Figure 8.** Blocking IL-13 does not reduce ASM mass and virus replication in neonatal, PVM-infected RAGE deficient mice. (**A**) Schematic representation of the experimental protocol. Neonatal RAGE deficient mice were infected with 10 PFU of PVM at seven days of age and treated i.p. with 100 μg of sIL-13Ra2-Fc or an IgG isotype control once per day from two to six days post-infection. (**B**) Percentage of airway epithelial cells (AECS) positive for PVM in treated mice seven days post-PVM infection (i.e. at 14-days of age). (**C**) The percentage of airway epithelial cells (AECs) with cytoplasmic HMGB1 in treated mice seven days post-PVM infection. (**D**) Lung sections were obtained from treated mice seven days post-PVM infection (i.e. at 14-days of age). These were then stained for smooth muscle actin by immunohistochemistry and the airway smooth muscle area (ASM) was calculated relative to the basement membrane (BM) length of small airways.). Statistical significance was determined by a Student's t-test. Statistical significance is denoted by asterisks (*p<0.05;**p<0.01; ***p<0.001; ns: not significant). Box and whisker plots show the minimum value, the median and the maximum value. Data are derived from a single experiment. n = 5–6.

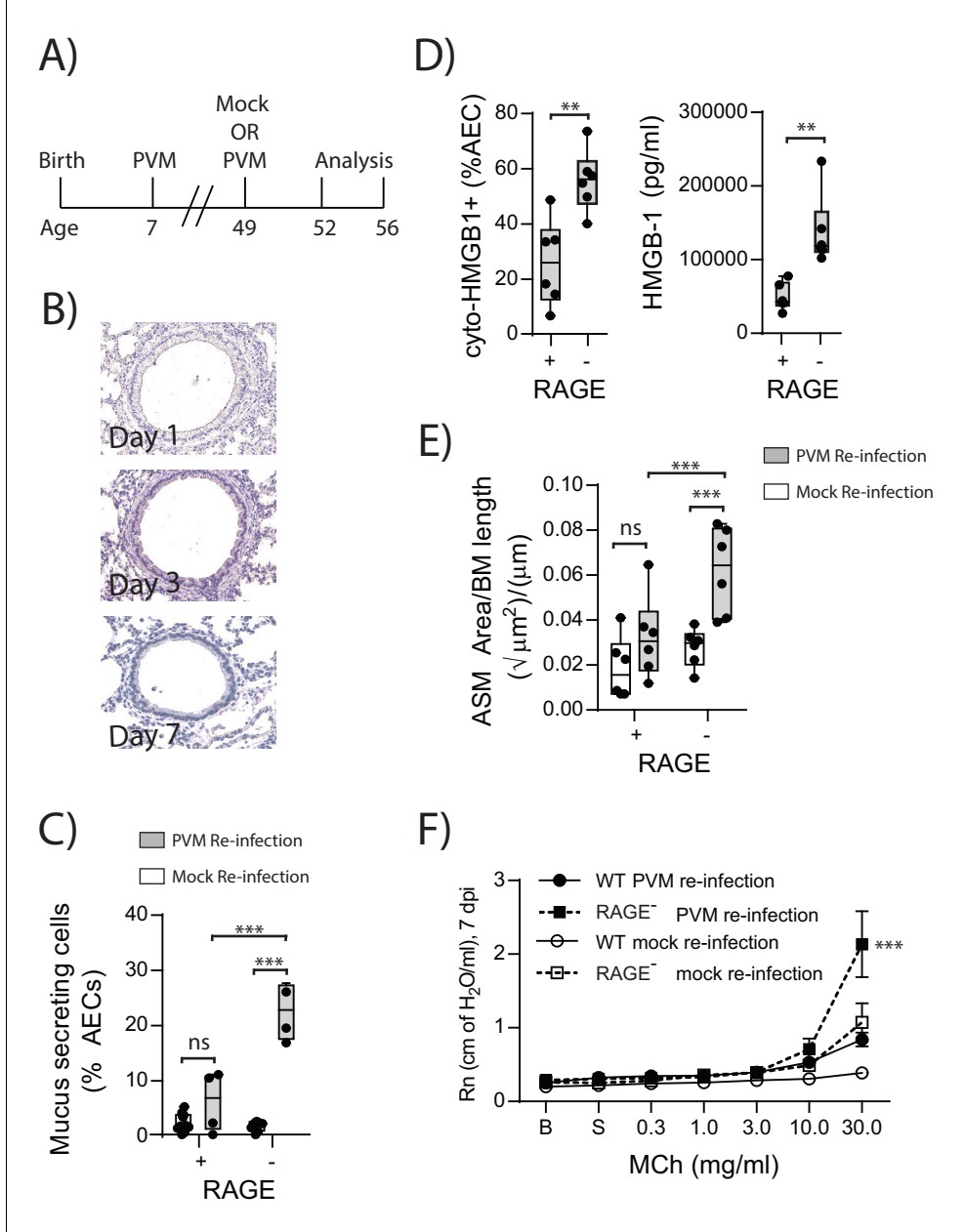

**Figure 9.** Reinfection with PVM induces the cardinal features of asthma in adult RAGE deficient mice. (A) Schematic representation of the experimental protocol. Neonatal WT (RAGE+) and RAGE deficient mice were inoculated with PVM (10 PFU) or mock at seven days of age and later re-infected with PVM (100 PFU) or mock 42 days after the primary infection. (B) Representative images of PVM staining in the lungs of RAGE deficient mice at various timepoints post-infection. (C) Percentage of airway epithelial cells (AECs) producing mucus was quantified by immunohistochemistry at seven days post-reinfection (i.e. at 56-days of age). (D) The percentage of airway epithelial cells (AECs) with cytoplasmic HMGB1 three days post re-infection (i.e. at 52-days of age). Levels of extracellular HMGB1 in the bronchoalveolar lavage fluid of mice three days post re-infection (i.e. at 52-days of age) are also displayed. (E) Lung sections from WT and RAGE deficient mice seven days post re-infection (i.e. at 56-days of age) were stained for smooth muscle actin by immunohistochemistry and the airway smooth muscle area (ASM) was calculated relative to the basement membrane (BM) length of small airways. (F) Airway hyperreactivity to increasing doses of aerosolised methacholine (MCh) at seven days post-reinfection (i.e. at 56-days of age). Statistical significance was determined by a two-way ANOVA with Tukey's multiple comparisons test (B, D and E) or a Student's t-test (C). Statistical significance is denoted by asterisks (*p<0.05;**p<0.01; ***p<0.001; ns: not

*Figure 9 continued on next page*

*Figure 9 continued*

significant). Box and whisker plots show the minimum value, the median and the maximum value. Data are representative of two independent experiments. n = 3–6.

different phenotypes of asthma. In the present study, we demonstrated impaired pDC recruitment and decreased production of IFN-α, -γ, -λ in RAGE deficient mice during an early life PVM infection, resulting in a higher viral load in the airway epithelium. The increased viral replication in neonatal RAGE deficient mice observed in this study was associated with the extracellular release of HMGB1, most likely as a consequence of virus-induced AEC stress. Increased levels of HMGB1 then drove an increase in ASM mass. This increase in ASM mass was dependent upon TLR4. Re-inoculation of RAGE deficient mice with PVM later in life, whilst resulting in a non-productive infection, exacerbated ASM growth and induced other characteristic features of asthma such as mucus hyper-secretion and airway hyperreactivity (*Figure 14*). Importantly, this asthma-like phenotype occurred in the absence of a pronounced eosinophilic or neutrophilic response. Thus, this model represents, to the best of our knowledge, the first mouse model that recapitulates the cardinal features of paucigranulocytic asthma.

In the present study anti-HMGB1 treatment in RAGE deficient mice reduced the levels of cytoplasmic HMGB1, which was associated with reduced airway remodelling (both in early and later life). The ability of anti-HMGB1 treatment to reduce cytoplasmic HMGB1 staining most likely reflects the ability of this antibody to inhibit extracellular HMGB1. Upon its release, HMGB1 can act back on cells to induce de novo HMGB1 synthesis and its release, to amplify the inflammatory response (*Porto et al., 2006*). Therefore, blocking extracellular HMGB1 can indirectly lower intracellular levels of HMGB1. Previous studies have also demonstrated that extracellular HMGB1 can be internalised (*Xu et al., 2014*). Thus, we cannot exclude the possibility that blocking extracellular HMGB1 reduced intracellular HMGB1 by decreasing HMGB1 uptake (*Xu et al., 2014*). However, it is important to note that HMGB1 internalisation is a RAGE-dependent process (*Xu et al., 2014*) and hence less likely to be relevant in RAGE deficient mice.

At present, it remains to be determined how HMGB1 induces ASM growth. HMGB1 can induce smooth muscle proliferation *in vitro* (*Porto et al., 2006*), suggesting that HMGB1 is capable of

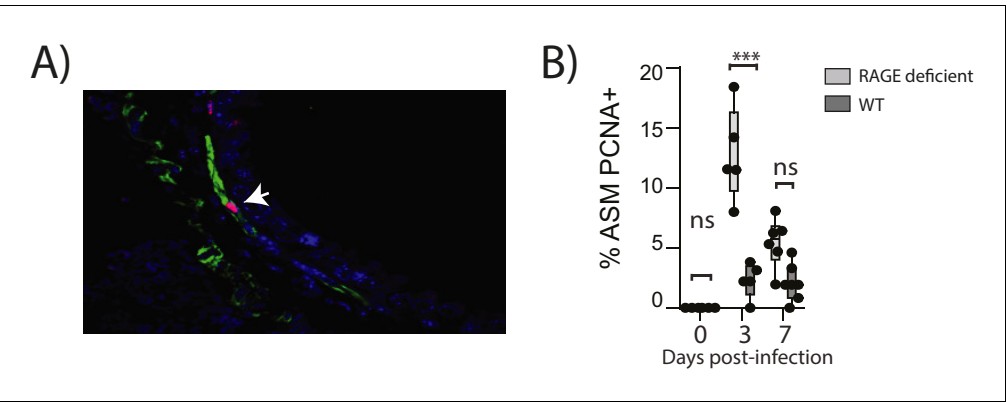

**Figure 10.** Reinfection with PVM induces airway smooth muscle proliferation in RAGE deficient mice. Neonatal mice were infected with PVM (10 PFU) at seven days of age and later re-infected with PVM (100 PFU) 42 days after the primary infection. (**A**) Representative immunofluorescent image (60X magnification) showing ASM cell proliferation in RAGE deficient mice three days post re-infection (i.e. at 52 days of age). ASM (green), PCNA (red) and DAPI (blue). Arrow indicates positive PCNA staining (**H**) The percentage of airway smooth muscle (ASM) cells positive for PCNA overtime. Statistical significance was determined by a two-way ANOVA with Tukey's multiple comparisons test. Statistical significance is denoted by asterisks (*p<0.05;**p<0.01; ***p<0.001; ns: not significant). Box and whisker plots show the minimum value, the median and the maximum value. Data are representative of two independent experiments. n = 5–7.

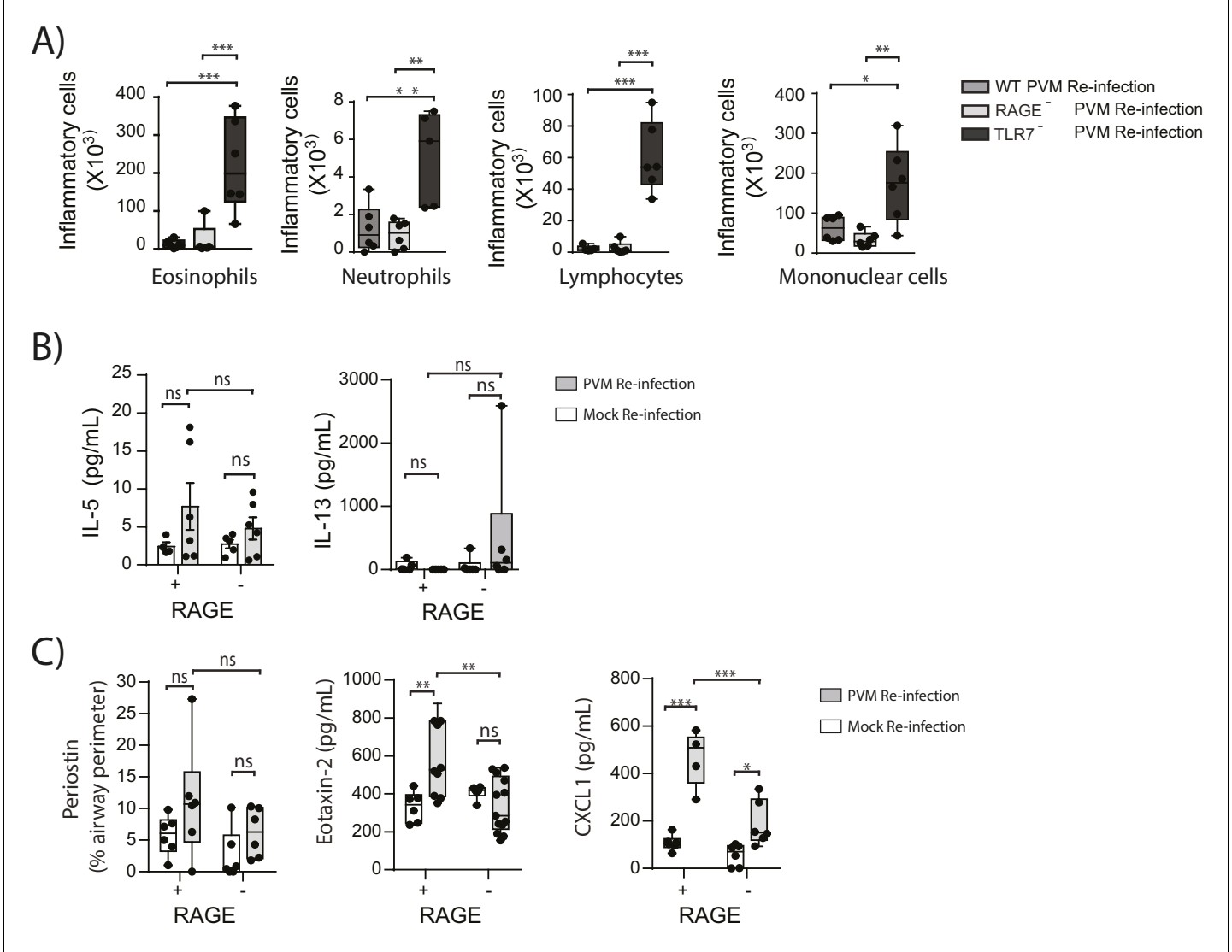

**Figure 11.** RAGE deficient mice do not display a pronounced T$_H$2 response upon re-infection with PVM. Neonatal WT (RAGE$^+$), TLR7 deficient and RAGE deficient mice were infected with PVM (10 PFU) or mock at seven days of age and later re-infected with PVM (100 PFU) or mock 42 days after the primary infection. (**A**) Inflammatory cells in the bronchoalveolar lavage fluid were enumerated by differential counting following Geimsa staining seven days post-reinfection (i.e. at 56-days of age). (**B**) Cytokine levels in the bronchoalveolar lavage fluid of neonatal mice seven days post-re infection (i.e. at 56-days of age). (**C**) Cytokine levels in the lungs of neonatal mice seven days post- reinfection (i.e. at 56-days of age). Periostin levels were determined by immunohistochemistry and quantified as a percentage of the epithelial basement membrane length. Statistical significance was determined by a Student's t-test (**A**) and a two-way ANOVA with Tukey's multiple comparisons test (**B** and **D**). Statistical significance is denoted by asterisks (*p<0.05; **p<0.01; ***p<0.001; ns: not significant). Box and whisker plots show the minimum value, the median and the maximum value. Data are representative of two independent experiments. n = 4–9.

mediating ASM growth in the absence of any other intermediary signalling molecule. Interestingly, blocking HMGB1 during the primary PVM infection also served to increase IFN–α and IFN-λ production and reduced viral load in RAGE deficient mice. We did not explore the mechanism by which anti-HMGB1 augments IFN production in our model. However, HMGB1 can inhibit IFN-α production by pDCs in response to CpG containing DNA (*Lotze and Tracey, 2005*), highlighting an important role for HMGB1 in regulating type I interferon responses.

The data presented herein raise the intriguing possibility that blocking HMGB1 may be a novel and effective treatment for early life viral infections and a possible preventative for asthma onset.

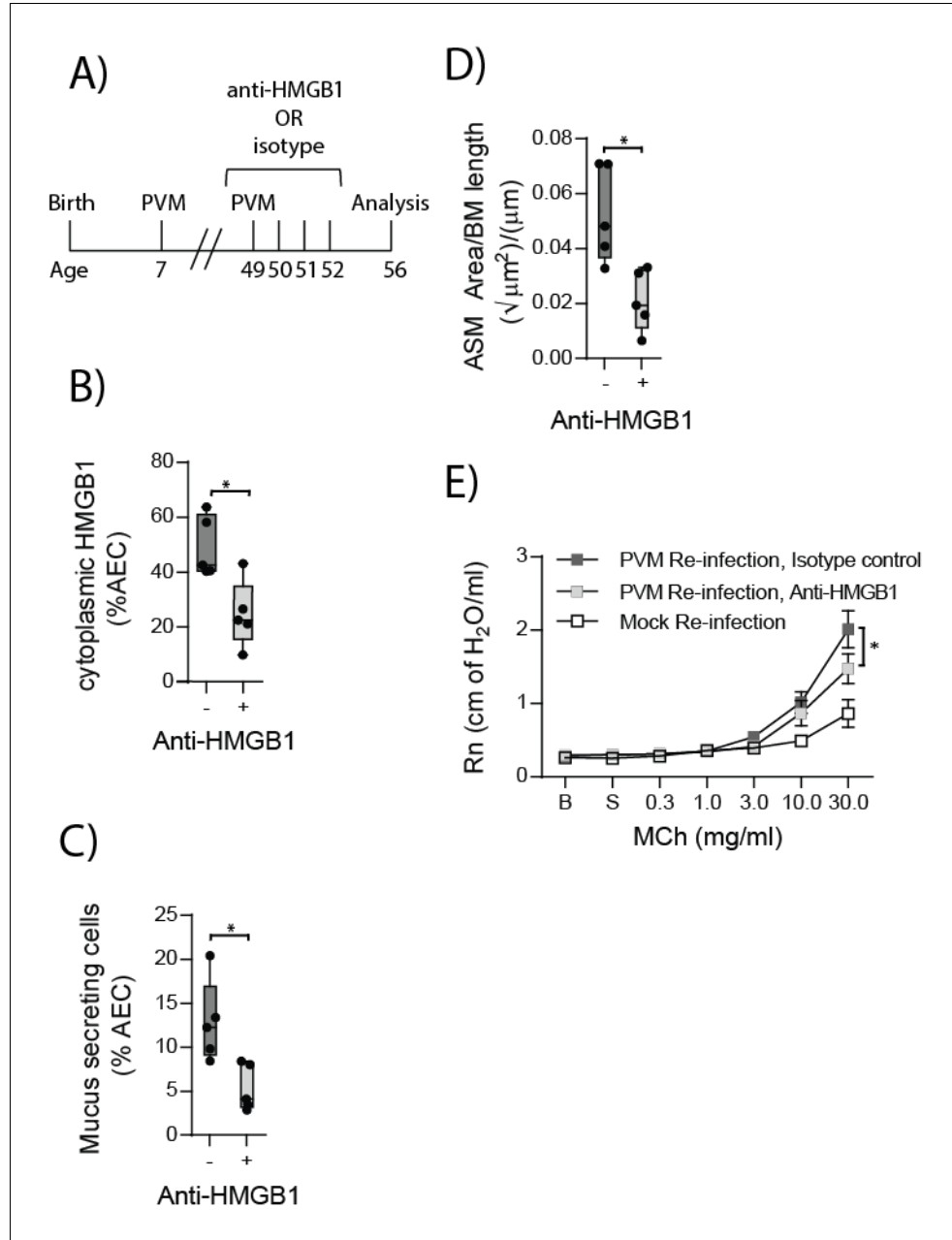

**Figure 12.** Anti-HMGB1 treatment following re-infection with PVM protects against asthma development in RAGE deficient mice. (**A**) Schematic representation of the experimental protocol. RAGE deficient mice were infected with 10 PFU of PVM at seven days of age and later re-infected with 100 PFU of PVM 42 days post-primary infection. RAGE deficient mice were treated once per day from day zero to day four post-reinfection with 50 µg of an anti-HMGB1 antibody or an IgG isotype control. (**B**) The percentage of airway epithelial cells (AECs) with cytoplasmic HMGB1 seven days post-PVM re-infection in treated mice (i.e. at 56-days of age). (**C**) Percentage of airway epithelial cells (AECs) producing mucus in the lungs of treated mice was quantified by immunohistochemistry at seven days post-reinfection (i.e. at 56-days of age). (**D**) Lung sections were obtained from treated mice seven days post re-infection (i.e. at 56-days of age). These were then stained for smooth muscle actin by immunohistochemistry and the airway smooth muscle area (ASM) was calculated relative to the basement membrane (BM) length of small airways. (**E**) Airway hyperreactivity to increasing doses of aerosolised methacholine (MCh) in treated mice at seven days post-reinfection (i.e. at 56-days of age). Statistical significance was determined by a Student's t-test (**A–C**) and a two-way ANOVA with Tukey's multiple comparisons test (**D**). Statistical significance is denoted by asterisks (*$p<0.05$;**$p<0.01$; ***$p<0.001$; ns: not significant). Box and whisker plots show the minimum value, the median and the maximum value. Data are derived from a single experiment. n = 5.

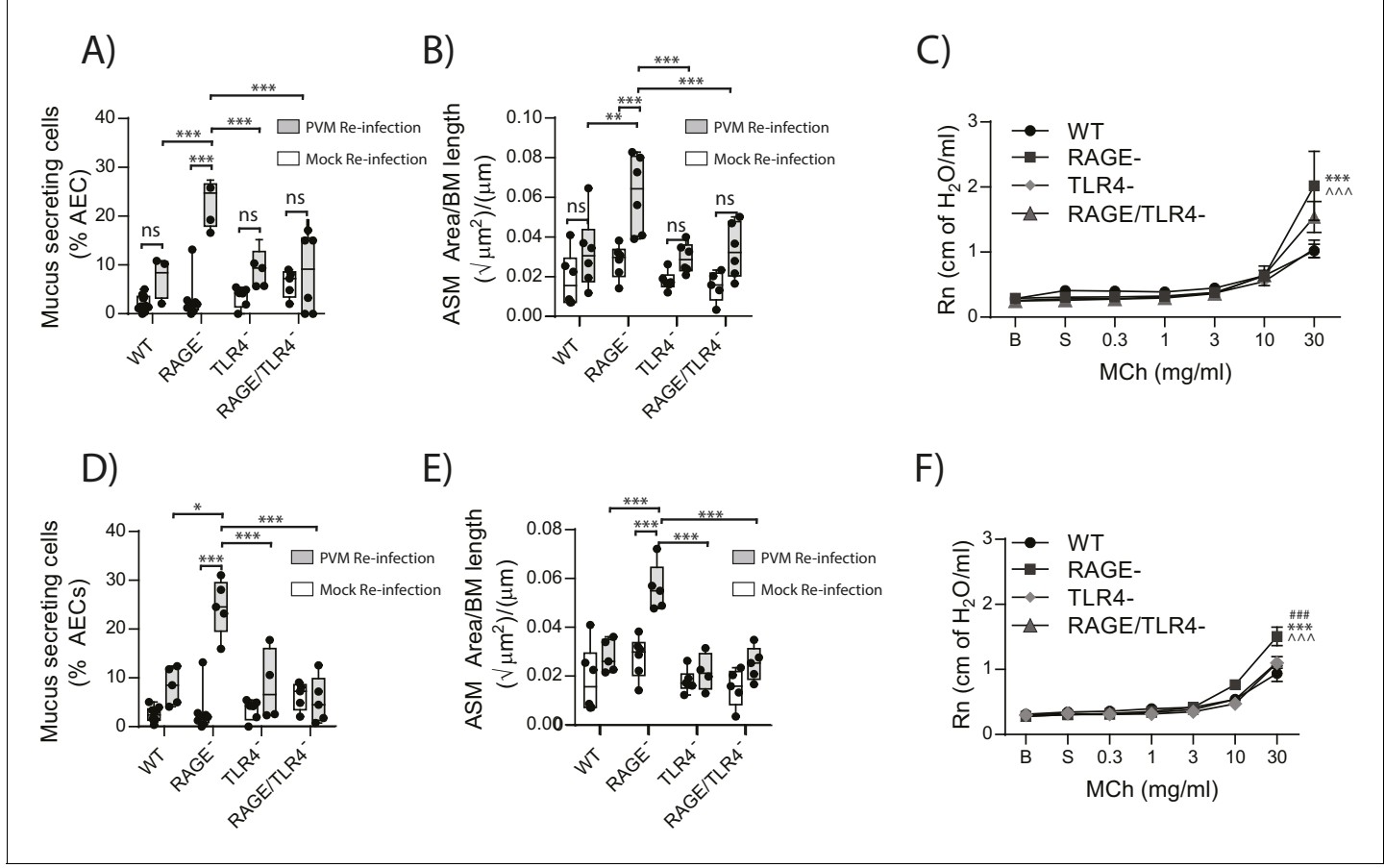

**Figure 13.** RAGE/TLR4 deficient mice are protected from asthma development following re-infection with PVM. Neonatal mice were infected with PVM (10 PFU) or mock at seven days of age and later re-infected with PVM (100 PFU) or mock 42 days after the primary infection. (**A**) Percentage of airway epithelial cells (AECs) producing mucus in the lungs of treated mice was quantified by immunohistochemistry at seven days post-reinfection (i.e. at 56-days of age) and (**D**) 21 days post-reinfection (i.e. at 70-days of age). (**B**) Lung sections were obtained from WT and RAGE deficient mice seven days post re-infection (i.e. at 56-days of age) and (**E**) 21 days post-reinfection (i.e. at 70-days of age). These were then stained for smooth muscle actin by immunohistochemistry and the airway smooth muscle area (ASM) was calculated relative to the basement membrane (BM) length of small airways. (**C**) Airway hyperreactivity to increasing doses of aerosolised methacholine (MCh) in treated mice at seven days post-reinfection (i.e. at 56-days of age). Statistical significance is denoted by asterisks (one symbol p<0.05;two symbols p<0.01; three symbols p<0.001; ns: not significant). Statistical significance is shown relative to WT mice (*) and TLR4 deficient (^) mice. (**F**) Airway hyperreactivity to increasing doses of aerosolised methacholine (MCh) in treated mice at 21 days post-reinfection (i.e. at 70-days of age). Statistical significance was determined by a two-way ANOVA with Tukey's multiple comparisons test. Statistical significance is denoted by asterisks (one symbol p<0.05;two symbols p<0.01; three symbols p<0.001; ns: not significant). Statistical significance is shown relative to WT mice (*), TLR4 deficient (^) and RAGE/TLR4 deficient mice (#). Box and whisker plots show the minimum value, the median and the maximum value. Data are representative of two independent experiments. n = 3–6.

Although this hypothesis requires further investigation, blocking HMGB1 could be an approach that is readily translated to the clinic given that anti-HMGB1 treatments have already been approved for clinical trials to block systemic inflammation (*Ulloa and Messmer, 2006*). Interestingly, several of the viruses and bacteria that have been implicated in the pathogenesis of asthma are also known to induce HMGB1 (*Moisy et al., 2012*; *Hou et al., 2014*; *Wang et al., 2004*, *1999*). Indeed, our preliminary data suggest that later life exposure to either influenza A virus or lipopolysaccharide (known HMGB1 stimuli) (*Nosaka et al., 2015*; *Gardella et al., 2002*) are able to induce at least some features of airway remodelling in RAGE deficient mice that were infected with PVM in early life (data not shown). These data thus also raise the possibility that the mechanisms of disease shown here may not be limited to PVM-induced asthma. Similarly, we previously identified that HMGB1 was common to the pathogenesis of house dust mite- and cockroach-induced allergic asthma in mice

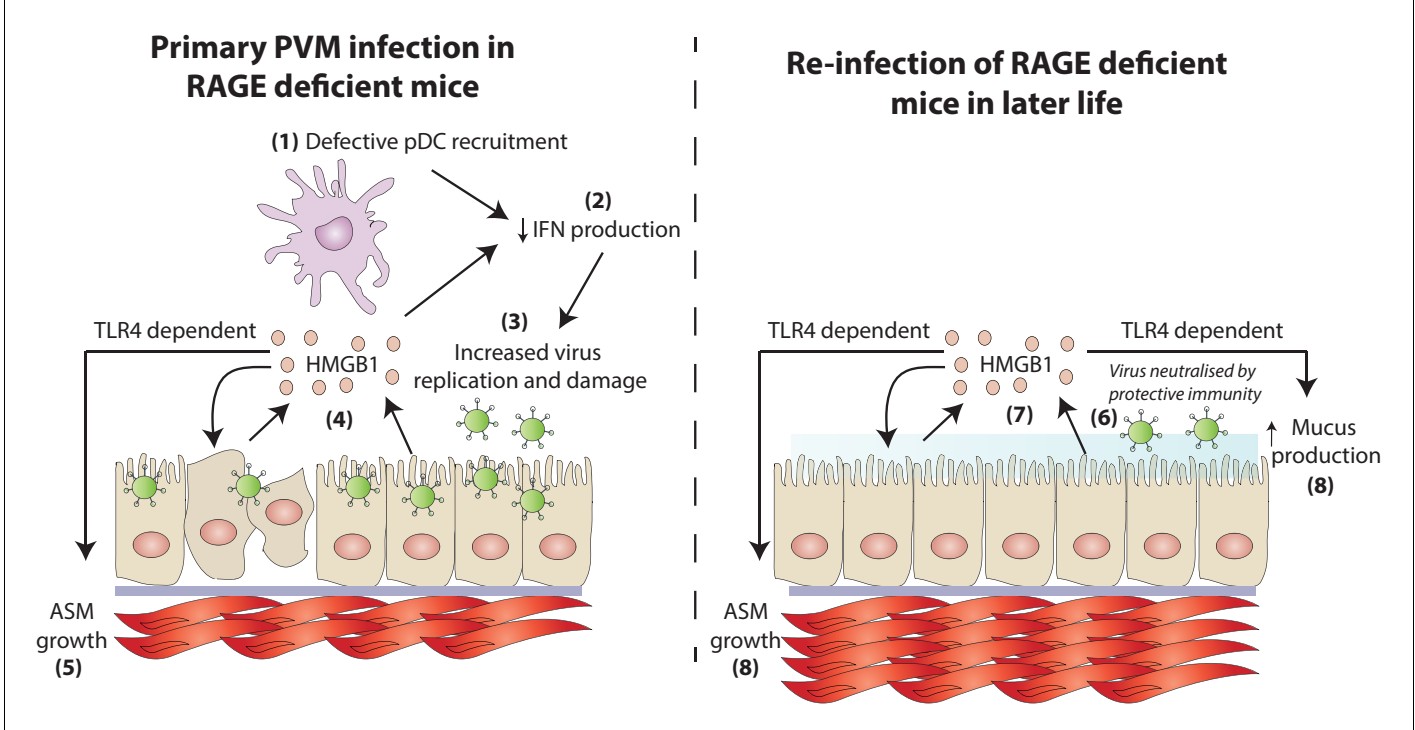

**Figure 14.** A schematic representation of the proposed model of disease. During a primary PVM infection, reduced interferon production results in increased virus replication. This is then associated with the release of HMGB1 (assumedly due to virus-induced stress/damage). In a TLR4-dependent fashion, HMGB1 drives the growth of airway smooth muscle (ASM). Upon reinfection with PVM later in life, there is no active virus replication (presumably due to a protective immune response). However, exposure to PVM triggers the production of HMGB1 which then drives ASM growth and mucus production in a TLR4 dependent manner.

(*Ullah et al., 2014*). Collectively, these findings and others highlight a central role for HMGB1 in the pathogenesis of multiple subtypes of asthma.

The redox state of HMGB1 controls which host receptors it will interact with. Specifically, when the cysteine in location C106 is in the reduced thiol form and C23 and C45 are involved in a disulphide bridge (HMGB1$^{C23–C45C106h}$, disulphide HMGB1), HMGB1 functions as a TLR4 ligand (*Yang et al., 2012*, *2010*). In light of the fact that TLR4 was essential for ASM growth and the cardinal features of asthma in this model, it is tempting to speculate that disulphide HMGB1/TLR4 interactions are sufficient to induce early life airway remodelling and the development of asthma later in life. This is consistent with the fact that the protective anti-HMGB1 antibody used in the present study blocks all-thiol and disulphide HMGB1 (*Yang et al., 2010*, *2012*; *Zhu et al., 2015*). Thus, the specific role of disulphide HMGB1 in the development of asthma represents an area of ongoing investigation.

Whilst the absence of both TLR7 and RAGE mediate asthma via similar mechanisms (i.e. a defective interferon response and an increase in ASM mass), this study showed that PVM-infected RAGE deficient mice displayed a markedly different inflammatory profile to that which we previously described in PVM-infected TLR7 deficient mice (*Kaiko et al., 2013*). Compared to TLR7 deficient mice, re-infected RAGE deficient mice did not develop type-2 eosinophilic inflammation nor did they generate a neutrophilic response. In TLR7 deficient mice, the type-2 inflammation was underpinned by an IFN-α$^{low}$/IL-33$^{high}$ cytokine microenvironment (*Kaiko et al., 2013*), whereas the absence of RAGE attenuated the production of both IFN–α and IL-33. Others have demonstrated that RAGE deficient mice produce lower levels of both type-1 and type-2 cytokines in an ovalbumin model of allergic asthma (*Akirav et al., 2014*). Similarly, we have previously shown that type-2 eosinophilic inflammation, induced in response to either house dust mite or cockroach sensitisation/challenge, is significantly reduced in the absence of RAGE (*Ullah et al., 2014*). The mechanism by which

the activation of RAGE contributes to type-2 inflammation remains poorly defined. However, it is noteworthy that RAGE is necessary for the lung infiltration of type-2 innate lymphoid cells (*Oczypok et al., 2015*), a cell type that is increasingly implicated in the priming and amplification of type two immunity (*Oliphant et al., 2014*). Similarly, the absence of a pronounced neutrophilic response in re-infected RAGE deficient mice is consistent with the role of RAGE in neutrophil recruitment (*Orlova et al., 2007*). Thus, although HMGB1-mediated activation of RAGE promotes type-2 inflammation (which is known to induce the hallmark pathologies of asthma), in the absence of RAGE HMGB1 can directly induce features of airway remodelling and thus contribute to airway hyperreactivity independently of type-2 inflammation.

Paucigranulocytic asthma is a poorly understood subtype of asthma. The mechanism of airway hyperreactivity and remodelling in these patients remains unclear, and indeed HMGB1 levels, although known to be elevated in eosinophilic asthma (*Watanabe et al., 2011*) have never been thoroughly investigated in in paucigranulocytic asthma. Moreover, whether specific genetic mutations lead to a propensity to develop pauciogranulocytic asthma (instead of, for example, eosinophilic asthma) has yet to be studied. However, the data presented here suggest that defective RAGE signaling contributes to the development and progression of airway remodeling in the absence of Th2/Th17-mediated granulocytic inflammation Given that we identified HMGB1 as a central mediator in the pathogenesis of airway remodelling, our findings suggest that anti-HMGB1 therapy may be a novel therapeutic for the treatment of asthma, irrespective of inflammatory cell subtype.

# Materials and methods

## Mice strains

C57BL/6 wild-type (WT) mice were purchased from the University of Queensland (Brisbane, Australia). RAGE-deficient mice were kindly provided by Prof. Ann-Marie Schmidt (New York University Langone Medical Centre, USA). TLR4 deficient mice were provided by Dr. Matthew Sweet (Institute for Molecular Bioscience, University of Queensland, Australia). TLR4 deficient mice were cross-bred with RAGE deficient mice to generate RAGE/TLR4 deficient mice. TLR7 deficient mice were provided by Prof. John Hayball (University of South Australia, Adelaide). All mice were on a C57BL/6 background, and rederived into a SPF facility prior to experimentation. Mice were housed in individually ventilated cages, on a 12 hr controlled day/night cycle with food and water available *ad libitum*. Dams were monitored daily from day 14 of gestation for litter birth. At 3 days of age, litter size of 6–8 pups per experimental group was standardised. The study was approved by the Animals Ethics Committee, University of Queensland (Ethics number 209/13).

## Virus strain

Stocks of PVM strain J3666 were prepared from mouse lung homogenates as described previously (*Garvey et al., 2005*).

## Infection and treatment of mice

Viral inoculums were prepared in Dulbecco's Modified Eagle Medium (DMEM, Invitrogen, Carlsbad, U.S.A.) containing 10% heat-inactivated Foetal Bovine Serum (FBS) (10% v/v) (Sigma-Aldrich, St Louis, U.S.A.). Diluent alone was used for mock infections. The response to PVM in neonatal mice was assessed by infecting seven-day old mice intranasally with 10 plaque forming units (PFU) of PVM or mock in a volume of 10 µLs. Where relevant, 42-days after the primary infection, mice were re-infected intranasally with 100 PFU of PVM or mock in a total volume of 50 µL. All intranasal infections were administered under isofluorane-induced anaesthesia. Where indicated, mice were treated intranasally with 5000U of mouse IFN-α (HyCult Biotechnology, Uden, the Netherlands), PBS, 50 µg of anti-HMGB1 antibody (kindly provided by Prof. Kevin J Tracey, Feinstein Institute of Medical Research, NY, USA) or an IgG isotype (eBioscience, CA, USA). The selected dose of IFN-α was based upon the findings of previous studies (*Nakajima et al., 1994*).

## Flow cytometry

Lungs were mechanically dissociated using a syringe plunger and 70 µm nylon cell strainer and single-cell suspensions were prepared. Alternatively, bone marrow-derived pDCs grown in culture were washed and resuspended in FACS buffer (PBS/2%FBS). Cells were incubated with 10 ng/ml of FcγRIII/II (Fc block; BD Biosciences) then stained for CD45RA (PE; clone 14.8, BD Biosciences), CD11c (V450; clone HL3, BD Biosciences) CD11b (V500, clone M1/70, BD Biosciences), Siglec H (APC, clone eBio440c, eBioscience) and B220 (PerCP-Cy5.5, clone RA3-6B2, eBioscience). Cells were fixed in formalin (0.5% v/v) and acquired using an LSRII flow cytometer (BD Bioscience).

## Bronchoalveolar lavage (BAL) collection and analysis of inflammatory cells

Lung lobes were lavaged with 400 µl (for neonatal mice) or 600 µl (for adult mice) of ice-cold PBS. Cells were pelleted by centrifugation and treated with red blood cell (RBC) lysis buffer. A StatSpin cytofuge was used at 600 rpm for 7 min to achieve a monolayer of leukocytes on triethoxysilane (Sigma-Aldrich) treated slides. 24 hr later the cells were fixed in 100 % v/v methanol for 15 min and stored until histological staining was ready to be performed. The slides were stained with May-Grunwald Giemsa solution (Sigma-Aldrich). Differential counts were performed using standard morphological criteria to classify cells as macrophages, lymphocytes, neutrophils and eosinophils.

## Immunohistochemistry

The superior right lung lobe was fixed in 10% neutral buffered formalin, embedded in paraffin, sectioned at 5 µm and subsequently used for immunohistochemistry. Antigen retrieval was performed by boiling the slides in 10 mM citrate buffer (pH 6) in a pressure cooker for 10 min. Sections were permeabilised using 0.5% Triton or 0.6% Tween-20 and blocked with 10% normal goat serum (Sigma-Aldrich). Slides were incubated with the relevant primary antibody overnight at 4°C (see *Table 1*). The relevant secondary antibody (Goat anti-rabbit-AP [Sigma-Aldrich], Goat anti-mouse AP [Sigma-Aldrich] or Streptavidin-AP [GE Healthcare Biosciences, NJ, U.S.A.]) was added to sections for 1 hr at room temperature and colour was developed using the Fast Red substrate (Sigma-Aldrich). Sections were counterstained with haematoxylin and mounted with Glycergel (Dako, CA, U.S.A.). Viral load was assessed as the percentage of airway epithelial cells positive for PVM on the apical surface and in the cytoplasm. Cytoplasmic HMGB1 was quantified as percentage of airway epithelial cells positive for HMGB1 in the cytoplasm. ASM was quantified essentially as described previously (*Lynch et al., 2016*) and was calculated as the $\sqrt{area}$ (µm²)/ basement membrane length (µm) of small airways (airway circumference <500 µm quantified for neonates and <800 µm for adult mice). To identify mucus producing cells, sections were stained with Periodic Acid-Schiff (PAS) and then counterstained with Harris' hematoxylin (Dako) and mounted in DEPEX. Mucus production was scored based on the percentage of mucus secreting AECs relative to the total number of AECs. At least five airways were quantified per sample. All stained slides were scanned using a digital slide scanner (Scanscope XT, Aperio Technologies, CA, USA) at 200X magnification. Photomicrographs of stained sections were taken at 20, 40 and 100x magnification using an Olympus BX51 microscope.

**Table 1.** Primary antibodies.

| Primary antibody (dilution) | Clone | Supplier |
| --- | --- | --- |
| Rabbit anti-PVM glycoprotein (G) antiserum (1:8000) | N/A | Gift from Dr. Ursula Buchholz, NIAID, USA |
| Rabbit anti-HMGB1 (1:400) | Ab 18256 | Abcam |
| Mouse anti-actinα-smooth muscle (1:400) | 1 A4 | Sigma-Aldrich |
| Biotinylated anti-periostin (1:400) | AG-20B-0033B-C100 | AdipoGen |

## Immunofluorescence

Lung sections were permeabilised with 0.5% Triton, blocked with 10% normal goat serum (Sigma-Aldrich) and incubated overnight at 4°C with anti-actin α-smooth muscle (1:400; Sigma-Aldrich) and biotinylated mouse anti-PCNA (1:200; Zymed Laboratories, CA, U.S.A.). Sections were then stained with goat anti-mouse AF488 (1:500; Invitrogen) and streptavidin AF647 (1:500; Invitrogen). Slides were subsequently counterstained in 4', 6-amino-2-phenylindole DAPI (final concentration: 0.5 μg/ml) for five mins. Photomicrographs of airways were taken at 40X magnification using a fluorescent microscope and the percentage of ASM cells positive for PCNA around the airway epithelial cells were quantified from four airways per mouse.

## Enzyme linked immunosorbent assay (ELISA)

Whole lung tissue was stored in radio immunoprecipitation (RIPA) buffer and homogenised using tissue tearer homogeniser (BioSpec Products Inc, OK, U.S.A). Alternatively, BAL fluid was collected and stored at −20°C until analysis. Cytokine levels were measured according to the manufacturer's protocol (Biolegend, CA, U.S.A; Chondex, WA, U.S.A; R and D systems, MN, U.S.A.; PBL Assay Science,Piscataway, NJ, U.S.A.). Detection limits were as follows: HMGB1 (1.6 ng/mL), IL-5 (4 pg/mL), IL-17A (2 pg/mL), Eotaxin-2 (15.6), CXCL1 (7.8 pg/mL), IL-12p40 (7.8 pg/mL), IL-1$\beta$ (7.8 pg/mL), IL-6 (2 pg/mL), IL-23 (8 pg/mL), CCL3 (7.8 pg/mL), TNF$\alpha$ (3.9 pg/mL), IFN-$\lambda$ (4 og/mL), IFN-$\alpha$ (5 pg/mL) and IFN-$\gamma$ (4 pg/mL).

## Cytokine bead analysis

Levels of IL-13 in BAL fluid were measured using a mouse IL-13 Flex Set (BD Biosciences; Detection limit: 15.3 pg/mL).

## Quantitative polymerase chain reaction (qPCR)

RNA was isolated from lung tissues by homogenisation with a sterile plastic pestle in 200 μl of TRI-reagent (Ambion, TX, U.S.A.) according to the manufacturer's protocol. DNase digestion was performed using TURBO DNA-free™ kit (Ambion). cDNA was generated from 1 μg of RNA using Superscript Reverse Transcriptase (Invitrogen). Real time PCR was performed on generated cDNA with SYBER Green (Invitrogen) using an ABI7900 system (Applied Biosystems, Victoria, Australia). Forward and reverse primer sequences for each gene of interest are shown in *Table 2*. Gene expression was normalised relative to hypoxanthine phosphoribosyl transferase (HPRT) expression and fold change was calculated using the $\Delta\Delta$Ct method. PVM titres were analysed via qPCR as described previously (*Davidson et al., 2011*).

**Table 2.** Primers used in the present study.

| Primer | Sequence |
|---|---|
| *IFN-$\alpha$* | FW: accaacagatccagaaggctcaag<br>RV: agtcttcctgggtcagaggaggtt |
| *IFN-$\beta$* | FW: agagttacactgcctttgccatcc<br>RV: ccacgtcaatctttcctcttgctt |
| *IFN-$\gamma$* | FW: tcttgaaagacaatcaggccatcc<br>RV: gaatcagcagcgactccttttcc |
| *IFN-$\lambda$* | FW: gattgccacattgctcagttc<br>RV: cttctcaagcagcctcttctc |
| *IRF-7* | FW: cttagccgggagcttggatctact<br>RV: cccttgtacatgatggtcacatcc |
| *HPRT* | FW: aggccagactttgttggatttgaa<br>RV: caacttgcgctcatcttaggcttt |
| *TNF-$\alpha$* | FW: gtctactgaacttcggggtgatcg<br>RV: agccttgtcccttgaagagaacct |
| PVM (*SH* domain) | FW: gcctgcatcaacacagtgtgt<br>RV: gcctgatgtggcagtgctt |

## Airway resistance

Airway resistance was measured by the forced oscillation technique (FOT) using a flexiVent FX machine (SCIEREQ, Quebec, Canada). Briefly, mice were anaesthetised and then mechanically ventilated. Increasing doses of nebulised methacholine (Sigma-Aldrich) were administered via the trachea at concentrations of 0, 0.3, 1, 3, 10 mg/ml. The response to methacholine was expressed as a percentage change over the baseline control (saline).

## Statistics

Graph Pad Prism software (version 6.00) was used for all statistical calculations. Data sets were analysed using unpaired Student's t-test, 1 way or 2-way ANOVA with Tukey's multiple comparison test. Outliers within data sets were excluded based on a Grubb's test/extreme studentised deviate (ESD) test for variation from a normal distribution.

## Study approval

All animal experiments were approved by The University of Queensland Animal Ethics and Experimentation Committee.

## Acknowledgements

This work was supported by an NHMRC project grant to S.P. and J.W.U. (1023756), an equipment grant (Rebecca L. Cooper Medical Research Foundation), an Australian Infectious Disease Research Excellence Award to S.P., an ARC Future Fellowship to S.P. and an NHMRC C.J. Martin post-doctoral fellowship to K.R.S. (1054081). We would like to thank Drs Nocka and Kasaian (Pfizer, Inc) for their gift of the sIL-13Ra2-Fc, Prof. Ann Marie Schmidt for her kind donation of RAGE deficient mice and Prof. Kevin J Tracey for his kind donation of the anti-HMGB1 antibody.

## Additional information

### Funding

| Funder | Grant reference number | Author |
| --- | --- | --- |
| National Health and Medical Research Council | 1054081 | Kirsty Renfree Short |
| National Health and Medical Research Council | 1023756 | John W Upham Simon Phipps |
| Australian Infectious Disease Research Excellence Award | | Simon Phipps |
| Australian Research Council | | Simon Phipps |
| Rebecca L. Cooper Medical Research Foundation | | Simon Phipps |

The funders had no role in study design, data collection and interpretation, or the decision to submit the work for publication.

### Author contributions

JA, Data curation, Formal analysis, Writing—original draft; MAU, WJG, ZL, RBW, JS, Data curation, Formal analysis, Writing—review and editing; KRS, Data curation, Formal analysis, Writing—original draft, Writing—review and editing; VZ, Data curation, Formal analysis, Methodology, Writing—review and editing; PDS, SBM, KMS, JWU, Conceptualization, Formal analysis, Supervision, Writing—review and editing; MARF, Conceptualization, Formal analysis, Writing—review and editing; MBS, Conceptualization, Formal analysis, Supervision, Writing—original draft, Writing—review and editing; SP, Conceptualization, Formal analysis, Supervision, Funding acquisition, Writing—original draft, Writing—review and editing

### Author ORCIDs

Simon Phipps, http://orcid.org/0000-0002-7388-3612

### Ethics

Animal experimentation: All animal experiments were approved by the University of Queensland Animal Ethics and Experimentation Committee (SBMS/209/13/NHMRC).

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
