## [Decision Letter]

Thank you for submitting your article "RAGE deficiency predisposes mice to virus-induced paucigranulocytic asthma" for consideration by *eLife*. Your article has been favorably evaluated by Wenhui Li (Senior Editor) and three reviewers, one of whom, Jos WM van der Meer (Reviewer #1), is a member of our Board of Reviewing Editors. The following individuals involved in review of your submission have agreed to reveal their identity: Sebastian Johnston (Reviewer #2); Jeff Whitsett (Reviewer #3).

The reviewers have discussed the reviews with one another and the Reviewing Editor has drafted this decision to help you prepare a revised submission.

Summary:

This is an interesting manuscript, reporting that RAGE deficiency pre-disposes mice to develop some features of asthma that do not include increased numbers of eosinophils and neutrophils, thus bearing some resemblance to a human phenotype of asthma which has been coined paucigranulocytic. Based on extensive and well-performed mouse studies, the paper elucidates pathways that are likely to be important in the human pathology. The paper opens up ways for a possible new treatment (anti HMGB1) for the human disease.

Essential revisions:

1) While paucigranulocytic asthma has been identified in quite a large number of studies now, in most it is not as common as the authors suggest, with a prevalence much lower than 40%. Also, tone down the references used to "paucigranulocytic asthma" because this is a concept that has really only been identified in a cross-sectional manner with no longitudinal demonstration that is a stable phenotype over time, and therefore a real stable asthma phenotype in man. It is interesting that the asthma end points that the authors studied in mice showed changes that are relevant to asthma; they find no increases in neutrophils and eosinophils and this does suggest that if paucigranulocytic asthma is a real stable phenotype, and that the mechanisms they have identified may be relevant to that phenotype. As mentioned, it is not yet accepted as a stable phenotype different from other phenotypes.

2) As said, most experiments are well executed, and in a series of sequential experiments the relevant pathways probed. Still the sequence of events is not completely clear. It would help if the authors would provide a scheme.

3) In line with the previous comment, the authors should comment on their finding that anti-HMGB1 also increased the IFN expression and reduced viral load (–subsection “Neutralising HMGB1 reduces PVM infection and ASM mass in RAGE^-/-^ mice during an early life PVM infection”). Do they think this is a positive feedback loop, or is HMGB1acting proximal to the IFN pathway? The administration of IFN α does not support the latter possibility (subsection “Administration of exogenous IFN-α limits viral replication, ASM remodelling and HMGB1 release in RAGE^-/-^ mice during an early life PVM infection”). At least this deserves more discussion.

4) The reinfection experiment is a bit puzzling: despite quite some protective immunity, a rather dramatic effect is seen. Do the authors have information whether dead virus (or more relevant an unrelated HMGB1 stimulus) would be able to do the same in RAGE^-/-^ mice?

5) It is unclear how many times experiments were repeated. It is a normal requirement that experiments are repeated at least twice to give a total of 3. Please state how many times each experiment was repeated in the legend, and if they have only been done once then they should be repeated.

6) Most end points are assessed only on day 7 with 1 or 2 additionally assessed on day 3. This is unusual with a virus infection which is a dynamic event and it is particularly unusual to present virus load data only on a single time point, as is done for the SH gene copy number. Please either provide a full-time course for the virus load data or reference previous work if it has been established by previous work that day 7 is the optimum day for assessment of the majority of outcomes.

7) The Discussion should not start with a new introduction, but rather present the major findings first. So delete the first paragraph, and rephrase the second paragraph.

---

## [Author Response]

*Essential revisions:*

*1) While paucigranulocytic asthma has been identified in quite a large number of studies now, in most it is not as common as the authors suggest, with a prevalence much lower than 40%. Also, tone down the references used to "paucigranulocytic asthma" because this is a concept that has really only been identified in a cross-sectional manner with no longitudinal demonstration that is a stable phenotype over time, and therefore a real stable asthma phenotype in man. It is interesting that the asthma end points that the authors studied in mice showed changes that are relevant to asthma; they find no increases in neutrophils and eosinophils and this does suggest that if paucigranulocytic asthma is a real stable phenotype, and that the mechanisms they have identified may be relevant to that phenotype. As mentioned, it is not yet accepted as a stable phenotype different from other phenotypes.*

The quoted statistic in regards the prevalence of paucigranulocytic asthma was considered to be representative of the disease prevalence found across multiple different studies (all of which are listed below). However, in response to the reviewer’s suggestion we have now replaced this average with the range derived from the 5 different studies listed below (see Introduction, first paragraph) and we have removed the prevalence estimates from the revised manuscript. In addition, we have also removed or modified the term paucigranulocytic asthma in the Abstract, Results and Discussion sections of the revised manuscript and acknowledged that there is limited information available regarding the stability of this phenotype overtime (Introduction, first paragraph).

StudyPrevalence of paucigranulocytic asthmaPopulation studiedSchleich et al.,BMC Pulmonary Medicine201340%508 patients with asthma recruited from the University Asthma Clinic of Liege.Simpson et al.,Respirology200632%93 subjects with a doctor’s diagnosis of asthma plus demonstrated AHR to hypertonic saline.Simpson et al.,Thorax200733%49 non-smoking subjects with a clinical diagnosis of symptomatic asthma and AHR to hypertonic salinePorsbjerg et al.,Journal of Asthma200931%62 asthmatic subjects with current asthma symptoms (past 4 weeks)Wang et al.,European Respiratory Journal201151.7%29 adults with stable asthmaWang et al.,European Respiratory Journal201149%49 children with stable asthma

*2) As said, most experiments are well executed, and in a series of sequential experiments the relevant pathways probed. Still the sequence of events is not completely clear. It would help if the authors would provide a scheme.*

In order to improve the clarity of the various experiments, we have now included schematic models of the experimental design in Figure 3, Figure 4, Figure 6, Figure 8, Figure 9 and Figure 12. We have also created a summary figure (Figure 14) to show the proposed model of disease.

*3) In line with the previous comment, the authors should comment on their finding that anti-HMGB1 also increased the IFN expression and reduced viral load (–subsection “Neutralising HMGB1 reduces PVM infection and ASM mass in RAGE^-/-^ mice during an early life PVM infection”). Do they think this is a positive feedback loop, or is HMGB1acting proximal to the IFN pathway? The administration of IFN α does not support the latter possibility (subsection “Administration of exogenous IFN-α limits viral replication, ASM remodelling and HMGB1 release in RAGE^-/-^ mice during an early life PVM infection”). At least this deserves more discussion.*

We hypothesise that RAGE signalling is required for recognition of viral RNA and the early induction of antiviral immunity. In the absence of viral control, the elevated viral load in the epithelium leads to the release of HMGB1. We hypothesise that the HMGB1 further blunts the antiviral host response by inhibiting type I IFN production by plasmacytoid dendritic cells, as described by Lotze and Tracey (Nat. Immunol. 2005). Inhibition of interferon production then results in increased viral replication which, in turn, results in increased HMGB1 release in a positive feedback loop. In response to the reviewer’s query we have now included a discussion of this point in the third paragraph of the Discussion. As described above, we have also created a summary figure (Figure 14) to show the proposed feedback loop.

*4) The reinfection experiment is a bit puzzling: despite quite some protective immunity, a rather dramatic effect is seen. Do the authors have information whether dead virus (or more relevant an unrelated HMGB1 stimulus) would be able to do the same in RAGE^-/-^ mice?*

We acknowledge the reviewer’s puzzlement, as we ourselves do not fully understand the mechanism by which the secondary infection induces the asthma-like features in the absence of a robust infection. Our working hypothesis is that the virus infects but does not replicate in macrophages or the airway epithelium and that the infection is sufficient to activate a pathway that induces the release of HMGB1. In addition to the experiments performed herein, we were also interested in determining whether the observed effects were restricted to reinfection with PVM. To answer this question we have performed experiments whereby we replaced the secondary infection with PVM with influenza A virus (PR8/34[H1N1]), a known stimulant of HMGB1 production (e.g. Nosaka et al., Critical Care, 2015). The data shown in Figure 15 demonstrate that influenza A virus (IAV) is also able to trigger ASM growth in RAGE^-/-^ mice that have been infected with PVM in early life. In an attempt to address the reviewers’ question, we also investigated whether LPS, an inducer of HMGB1 release (Gardella et al.,2002; EMBO reports) but also a ligand of TLR4, would be sufficient to promote airway remodelling. Mice were infected with PVM in early life were treated intranasally with LPS at 42-47 days post primary infection. LPS increased ASM levels above those observed in PVM + mock treated mice. How an early life infection with PVM alters the pulmonary environment such that the mice are ‘sensitised’ to a later life viral infection or pro-inflammatory stimulus remains an area of ongoing research in our laboratory. We have now included discussion of these data in the sixth paragraph of the Discussion.

Author response image 1.**DOI:**
http://dx.doi.org/10.7554/eLife.21199.018

*5) It is unclear how many times experiments were repeated. It is a normal requirement that experiments are repeated at least twice to give a total of 3. Please state how many times each experiment was repeated in the legend, and if they have only been done once then they should be repeated.*

Unfortunately, our animal ethics committee prohibits the repetition of in vivoexperiments three times in order to uphold the three Rs (replacement, reduction and refinement) underpinning the humane use of animals in scientific research. Thus, all experiments are representative of two independent experiments. The only exceptions to this are experiments where mice were treated with anti-HMGB1 and anti- sIL-13Ra2-Fc. This decision was made due to the limited availability of both the anti-HMGB1 and sIL-13Ra2-Fc (both were gifts from overseas researchers/companies and hence we only had a finite amount of material to work with). For transparency, all this information is now included in the relevant figure legends and alongside the relevant ‘n’ numbers.

*6) Most end points are assessed only on day 7 with 1 or 2 additionally assessed on day 3. This is unusual with a virus infection which is a dynamic event and it is particularly unusual to present virus load data only on a single time point, as is done for the SH gene copy number. Please either provide a full-time course for the virus load data or reference previous work if it has been established by previous work that day 7 is the optimum day for assessment of the majority of outcomes.*

Day 7 was selected as the time point of interest as we and others have previously established that this represents the peak of viral replication and clinical signs in mice infected with PVM (e.g. Bonville et al., 2006, Virology & Kaiko et al., 2013; JACI). We have now included these references in the first paragraph of the subsection “RAGE deficiency results in increased viral infection and reduced anti-viral immunity during a PVM infection in early life”.

7) The Discussion should not start with a new introduction, but rather present the major findings first. So delete the first paragraph, and rephrase the second paragraph.

In accordance with the reviewer’s suggestions we have now deleted the first paragraph and rephrased the start of the Discussion section.